# Negation mitigates rather than inverts the neural representations of adjectives

**Arianna Zuanazzi**[1]*, **Pablo Ripollés**[1,2,3], **Wy Ming Lin**[4], **Laura Gwilliams**[5], **Jean-Rémi King**[1,6°], **David Poeppel**[1,3,7°]

**1** Department of Psychology, New York University, New York, New York, United States of America, **2** Music and Audio Research Lab (MARL), New York University, New York, New York, United States of America, **3** Center for Language, Music and Emotion (ClaME), New York University, New York, New York, United States of America, **4** Hector Research Institute for Education Sciences and Psychology, University of Tübingen, Tübingen, Germany, **5** Department of Psychology, Stanford University, Stanford, California, United States of America, **6** Ecole Normale Supérieure, PSL University, Paris, France, **7** Ernst Strüngmann Institute for Neuroscience, Frankfurt, Germany

° These authors contributed equally to this work.
* az1864@nyu.edu

**Data Availability Statement:** All relevant data and codes of main data analyses are available on the Open Science Framework https://doi.org/10.17605/OSF.IO/5YS6B.

## Abstract

Combinatoric linguistic operations underpin human language processes, but how meaning is composed and refined in the mind of the reader is not well understood. We address this puzzle by exploiting the ubiquitous function of negation. We track the online effects of negation ("not") and intensifiers ("really") on the representation of scalar adjectives (e.g., "good") in parametrically designed behavioral and neurophysiological (MEG) experiments. The behavioral data show that participants first interpret negated adjectives as affirmative and later modify their interpretation towards, but never exactly as, the opposite meaning. Decoding analyses of neural activity further reveal significant above chance decoding accuracy for negated adjectives within 600 ms from adjective onset, suggesting that negation does not invert the representation of adjectives (i.e., "not bad" represented as "good"); furthermore, decoding accuracy for negated adjectives is found to be significantly lower than that for affirmative adjectives. Overall, these results suggest that negation mitigates rather than inverts the neural representations of adjectives. This putative suppression mechanism of negation is supported by increased synchronization of beta-band neural activity in sensorimotor areas. The analysis of negation provides a steppingstone to understand how the human brain represents changes of meaning over time.

## Introduction

A hallmark of language processing is that we combine elements of the stored inventory—informally speaking, words—and thereby flexibly generate new meanings or change current meanings. The final representations derive in systematic ways from the combination of individual pieces. The composed meanings can be extracted in relatively straightforward ways, such as by sequentially combining individual meanings of words and phrases (e.g., "this theory is correct") or stem from more subtle inferential processes, where further operations are required to achieve

**Funding:** This work was supported by the Leon Levy Foundation (https://www.leonlevyfoundation.org/leon-levy-fellowship-neuroscience/ to A.Z.), the European Union's Horizon 2020 research and innovation program under grant agreement No 660086 (https://rea.ec.europa.eu/funding-and-grants/horizon-europe-marie-sklodowska-curie-actions_en to J-R.K.), the Bettencourt-Schueller Foundation (https://www.fondationbs.org/en to J-R.K.), the Philippe Foundation (https://www.philippefoundation.org/ to J-R.K.), the FrontCog grant ANR-17-EURE-0017 (https://anr.fr/ProjetIA-17-EURE-0017 to J-R.K.), the National Science Foundation (https://www.nsf.gov/, grant 2043717, to D.P, P.R.), and the Ernst Struengmann Foundation (https://www.esi-frankfurt.de/ to D.P.). The funders had no role in study design, data collection and analysis, decision to publish, or preparation of the manuscript.

**Competing interests:** The authors have declared that no competing interests exist.

**Abbreviations:** AUC, area under the curve; BEM, Boundary Element Model; dSPM, dynamic statistical parametric mapping; MEG, magnetoencephalography; MRI, magnetic resonance image; PCA, principal component analysis.

understanding (e.g., "this theory is not even wrong," meaning "this theory is incoherent"). A mechanistic understanding of the underlying processes requires characterization of how meaning representations are constructed in real time. There has been steady progress and productive debate on syntactic structure building [1–6]. In contrast, how novel semantic configurations are represented over time is less widely investigated. In the experimental approach pursued here, we build on the existing literature on precisely controlled minimal linguistic environments [7,8]. We deploy a new, simple parametric experimental paradigm that capitalizes on the powerful role that negation plays in shaping semantic representations of words. While negation is undoubtfully a complex linguistic operation that can affect comprehension as a function of other linguistic factors (such as discourse and pragmatics [9–11]), our investigation specifically focuses on how negation operates in phrasal structures. Combining behavioral and neurophysiological data, we show how word meaning is (and is not) modulated in controlled contexts that contrast affirmative (e.g., "really good") and negated (e.g., "not good") phrases. The results identify models and mechanisms of how negation, a compelling window into semantic representation, operates in real time.

Negation is ubiquitous—and, therefore, interesting in its own right. Furthermore, it offers a compelling linguistic framework to understand how the human brain builds meaning through combinatoric processes. Intuitively, negated concepts (e.g., "not good") entertain some relation with the affirmative concept (e.g., "good") as well as their counterpart (e.g., "bad"). The function of negation in natural language has been a matter of longstanding debate among philosophers, psychologists, logicians, and linguists [12]. In spite of its intellectual history and relevance (interpreting negation was, famously, a point of debate between Bertrand Russell and Ludwig Wittgenstein), comparatively little research investigates the cognitive and neural mechanisms underpinning negation. Previous work shows that negated phrases/sentences are processed with more difficulty (slower, with more errors) than the affirmative counterparts, suggesting an asymmetry between negated and affirmative representations; furthermore, state-of-the-art artificial neural networks appear to be largely insensitive to the contextual impacts of negation [13–20]. This asymmetry motivates one fundamental question: *how* does negation operate?

Studies addressing this question suggest that negation operates as a suppression mechanism by reducing the extent of available information [21–23], either in 2 steps [18,24–28] or in 1 incremental step [12,29–31]; other studies demonstrate that negation is rapidly and dynamically integrated into meaning representations [10,32], even unconsciously [33]. Within the context of action representation (e.g., "cut," "wish"), previous research suggests that negation recruits general-purpose inhibitory and cognitive control systems [34–41].

While the majority of neuroimaging studies focused on how negation affects action representation, psycholinguistic research shows that scalar adjectives (e.g., "bad-good," "close-open," "empty-full") offer insight into how negation operates on semantic representations of single words. These studies provide behavioral evidence that negation can either *eliminate* the negated concept and convey the opposite meaning ("not good" = "bad") or *mitigate* the meaning of its antonym along a semantic continuum ("not good" = "less good," "average," or "somehow bad"; [11,12,42–44]). Thus, the system of polar opposites generated by scalar adjectives provides an especially useful testbed to investigate changes in representation of abstract concepts along a semantic scale (e.g., "bad" to "good"), as a function of negation (e.g., "bad" versus "not good").

Here, we capitalize on the semantic continuum offered by scalar adjectives to investigate *how* negation operates on the representation of abstract concepts (e.g., "bad" versus "good"). First, we track how negation affects semantic representations over time in a behavioral mouse tracking study (and a replication study; **Fig 1A**). Next, we use magnetoencephalography (MEG) and a decoding approach to track the evolution of neural representations of target adjectives in affirmative and negated phrases (**Fig 1B**). Mouse tracking and decoding approaches allow us to quantify and compare dynamic changes in participants' interpretations

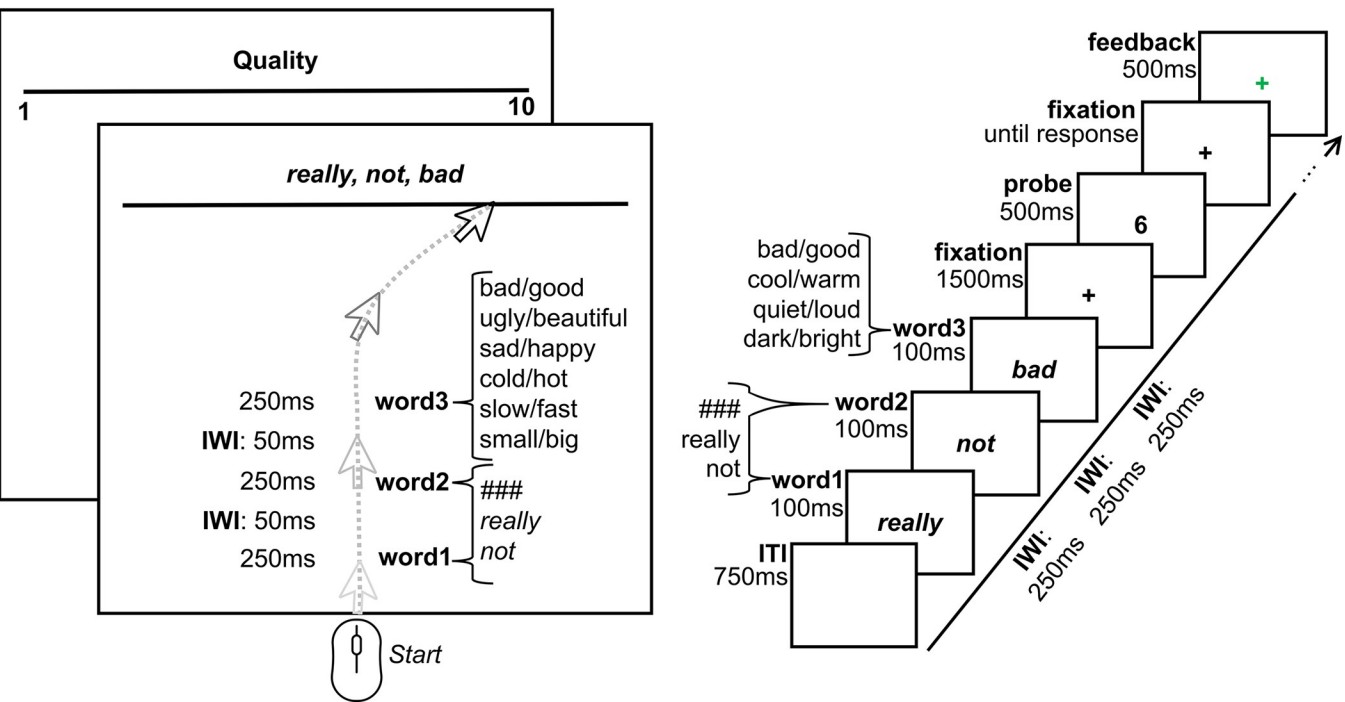

**Fig 1. Experimental procedures.** (**A**) Behavioral procedure. Participants read affirmative or negated adjective phrases (e.g., "really really good," "### not bad") word by word and rated the overall meaning of each phrase on a scale. Each trial consisted of combinations of "###," "really," and "not" in word positions 1 and 2, followed by an adjective representing the low or high pole across 6 possible scalar dimensions. Before each trial, participants were informed about the scale direction, e.g., "bad" to "good," i.e., 1 to 10. Scale direction was pseudorandomized across blocks. Feedback was provided at the end of each trial (to which 1 and 0 was assigned to compute the average feedback score). For each trial, we collected continuous mouse trajectories throughout the entire trial as well as reaction times. (**B**) MEG procedure. Participants read affirmative or negated adjective phrases and were instructed to derive the overall meaning of each adjective phrase on a scale from 0 to 8, e.g., from "really really bad" to "really really good." After each phrase, a probe (e.g., 6) was presented, and participants were required to indicate whether the probe number represented the overall meaning of the phrase on the scale (*yes/no* answer, using a keypad). Feedback was provided at the end of each trial (green or red cross, to which 1 and 0 was assigned to compute the average feedback score). While performing the task, participants lay supine in a magnetically shielded room while continuous MEG data were recorded through a 157-channel whole-head axial gradiometer system. (**A** and **B**) "###" = no modifier; IWI = inter-word-interval.

and neural representations of adjectives over time (e.g., [45,46]). We test 4 hypotheses: (1) negation does not change the representation of adjectives (e.g., "not good" = "good"); (2) negation weakens the representation of adjectives (e.g., "not good" < "good"); (3) negation inverts the representation of adjectives (e.g., "not good" = "bad"); and (4) negation changes the representation of adjectives to another representation (e.g., "not good" = e.g., "unacceptable"). The combined behavioral and neurophysiological data adjudicate among these hypotheses and identify potential mechanisms that underlie how negation functions in online meaning construction. Emerging temporal dynamics clarify how the effect of negation on adjective meaning unfolds over time, whether incrementally (i.e., in parallel to adjective processing) or serially (i.e., in a second step after adjective processing).

## Results

### Experiment 1: Continuous mouse tracking reveals a 2-stage representation of negated adjectives

Experiment 1 (online behavioral experiment; *N* = 78) aimed to track changes in representation over time of scalar adjectives in affirmative and negated phrases. Participants read 2-to-3-word

phrases comprising 1 or 2 modifiers ("not" and "really") and a scalar adjective (e.g., "really really good," "really not quiet," "not ### fast"). The number and position of modifiers were manipulated to allow for a characterization of negation in simple and complex phrasal contexts, above and beyond single word processing. Adjectives were selected to represent opposite poles (i.e., antonyms) of the respective semantic scales: *low* pole of the scale (e.g., "bad," "ugly," "sad," "cold," "slow," and "small") and *high* pole of the scale (e.g., "good," "beautiful," "happy," "hot," "fast," and "big"). A sequence of dashes was used to indicate the absence of a modifier. **Fig 1A** and **S1 Table** provide a comprehensive list of the linguistic stimuli. On every trial, participants rated the overall meaning of each phrase on a scale defined by each antonym pair (**Fig 1A**). Feedback was provided at the end of each trial (to which 1 and 0 were assigned to compute the average feedback score). We analyzed reaction times and continuous mouse trajectories, which consist of the positions of the participant's mouse cursor while rating the phrase meaning. Continuous mouse trajectories offer the opportunity to measure the unfolding of word and phrase comprehension over time, thus providing time-resolved dynamic data that reflect changes in meaning representation [15,45,47].

**Reaction times.** To evaluate the effect of antonyms and of negation on reaction times in behavioral Experiment 1, we performed a 2 (*antonym*: low versus high) × 2 (*negation*: negated versus affirmative) repeated-measures ANOVA. The results reveal a significant main effect of antonyms (F(1,77) = 60.83, $p < 0.001$, $\eta_p^2 = 0.44$) and a significant main effect of negation (F(1,77) = 104.21, $p < 0.001$, $\eta_p^2 = 0.57$, **Fig 2A**). No significant crossover interaction between antonyms and negation was observed ($p > 0.05$). Participants were faster for high adjectives (e.g., "good") than for low adjectives (e.g., "bad") and for affirmative phrases (e.g., "really really good") than for negated phrases (e.g., "really not good"). These results support previous behavioral data showing that negation is associated with increased processing difficulty [15,16]. A further analysis including the number of modifiers as factor (i.e., *complexity*) indicates that participants were faster for phrases with 2 modifiers, e.g., "not really," than phrases with one modifier, e.g., "not ###" (F(1,77) = 16.02, $p < 0.001$, $\eta_p^2 = 0.17$; see A Table in **S3 Table** for pairwise comparisons between each pair of modifiers), suggesting that the placeholder "###" may induce some processing slow-down. To confirm this hypothesis, further research should investigate the specific effect of placeholders (e.g., "###" or "xkq") on word and phrase representation and semantic composition.

**Continuous mouse trajectories.** Continuous mouse trajectories across all adjective pairs and across all participants are depicted in **Fig 2B** and **2C** (*low* and *high* summarize the 2 antonyms across all scalar dimensions; see **S1 Fig** for each adjective dimension separately).

To quantify how the final interpretation of scalar adjectives changes as a function of negation, we first performed a 2 (*antonym*: low versus high) × 2 (*negation*: negated versus affirmative) repeated-measures ANOVA for participants' ends of trajectories (filled circles in **Fig 2B**), which reveal a significant main effect of antonyms (F(1,77) = 338.57, $p < 0.001$, $\eta_p^2 = 0.83$), a significant main effect of negation (F(1,77) = 65.50, $p < 0.001$, $\eta_p^2 = 0.46$), and a significant antonyms by negation interaction (F(1,77) = 1346.07, $p < 0.001$, $\eta_p^2 = 0.95$). Post hoc tests show that the final interpretation of negated phrases is located at a more central portion on the semantic scale than that of affirmative phrases (affirmative low < negated high, and affirmative high > negated low, $p_{holm} < 0.001$). Furthermore, the final interpretation of negated phrases is significantly more variable (measured as standard deviations) than that of affirmative phrases (F(1,77) = 78.14, $p < 0.001$, $\eta_p^2 = 0.50$). Taken together, these results suggest that negation shifts the final interpretation of adjectives towards the antonyms, but never to a degree that overlaps with the interpretation of the affirmative antonym.

Second, we explored the temporal dynamics of adjective representation as a function of negation (i.e., from the presentation of word 1 to the final interpretation; lines in **Fig 2C**).

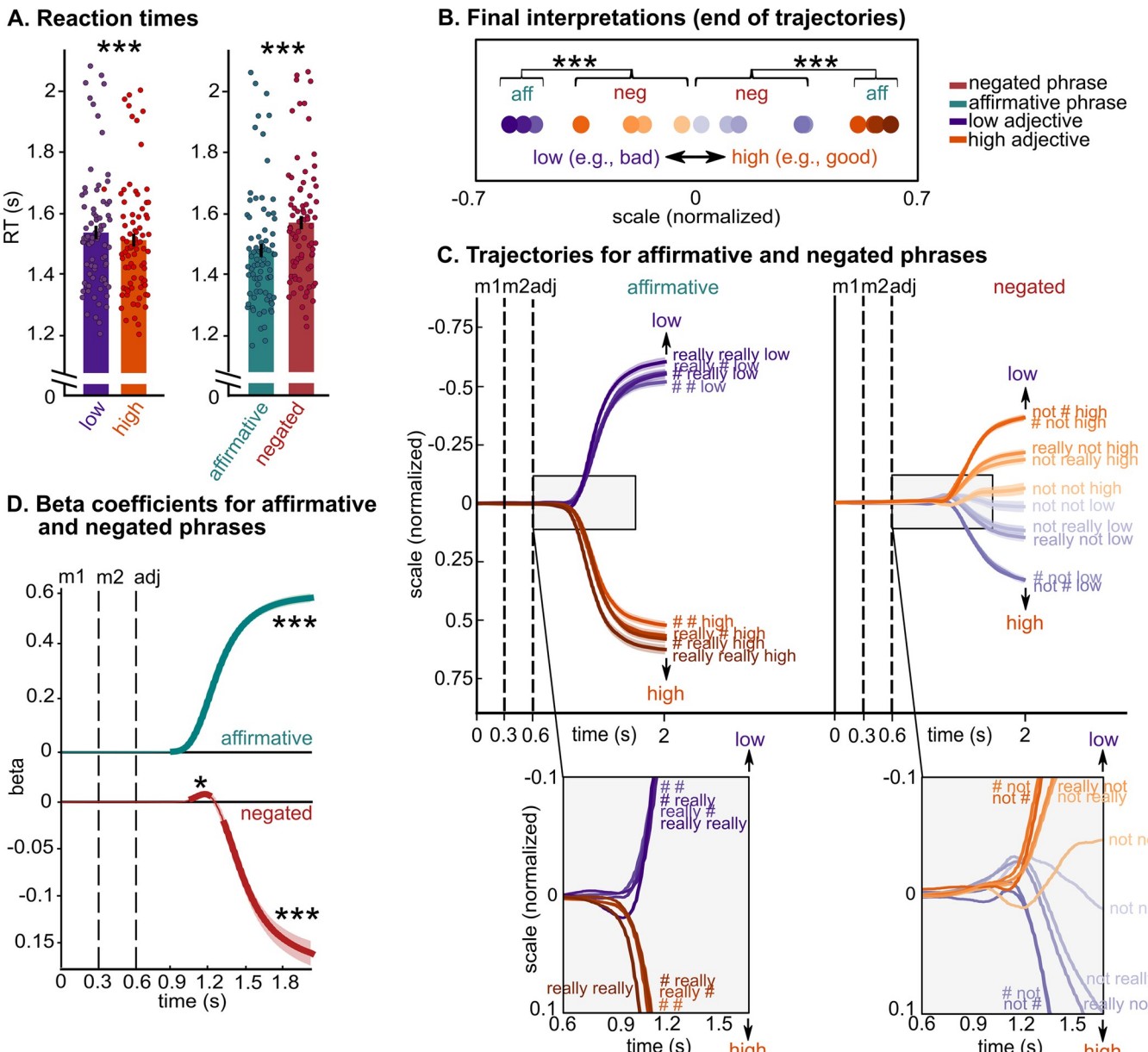

**Fig 2. Behavioral results. (A)** Reaction times results for the online behavioral study ($N = 78$). Bars represent the participants' mean ± SEM, and dots represent individual participants. Participants were faster for high adjectives (e.g., "good") than for low adjectives (e.g., "bad") and for affirmative phrases (e.g., "really really good") than for negated phrases (e.g., "really not good"). The results support previous behavioral data showing that negation is associated with increased processing difficulty. **(B)** Final interpretations (i.e., end of trajectories) of each phrase, represented by filled circles (purple = low, orange = high), averaged across adjective dimensions and participants, showing that negation never inverts the interpretation of adjectives to that of their antonyms. **(C)** Mouse trajectories for low (purple) and high (orange) antonyms, for each modifier (shades of orange and purple) and for affirmative (left panel) and negated (right panel) phrases. Zoomed-in panels at the bottom demonstrate that mouse trajectories of affirmative phrases branch towards the adjective's side of the scale and remain on that side until the final interpretation; in contrast, the trajectories of negated phrases first deviate towards the side of the adjective and subsequently towards the side of the antonym. This result is confirmed by linear models fitted to the data at each time point in **D**. **(D)** Beta values (average over 78 participants) over time, separately for affirmative and negated phrases. Thicker lines indicate significant time windows. **(C, D)** Black vertical dashed lines indicate the presentation onset of each word: modifier 1, modifier 2 and adjective; each line and shading represent participants' mean ± SEM. **(A, B, D)** *** $p < 0.001$; * $p < 0.05$. Data are available on the Open Science Framework https://doi.org/10.17605/OSF.IO/5YS6B.

While mouse trajectories of affirmative phrases branch towards either side of the scale and remain on that side until the final interpretation (lines in the left, gray, zoomed-in panel in **Fig 2C**), trajectories of negated phrases first deviate towards the side of the adjective and then towards the side of the antonym, to reach the final interpretation (i.e., "not low" first towards "low" and then towards "high"; right, gray, zoomed-in panel in **Fig 2C**; see **S1 Fig** for each adjective dimension separately). To characterize the degree of deviation towards each side of the scale, we performed regression analyses with antonyms as the predictor and mouse trajectories as the dependent variable (see **Materials and methods**). The results confirm this observation, showing that (1) in affirmative phrases, betas are positive (i.e., mouse trajectories moving towards the adjective) starting at 300 ms from adjective onset ($p < 0.001$, green line in **Fig 2D**); and that (2) in negated phrases, betas are positive between 450 and 580 ms from adjective onset (i.e., mouse trajectories moving towards the adjective, $p = 0.04$), and only become negative (i.e., mouse trajectories moving towards the antonym, $p < 0.001$) from 700 ms from adjective onset (red line in **Fig 2D**). Note that beta values of negated phrases are smaller than that for affirmative phrases, again suggesting that negation does not invert the interpretation of the adjective to that of the antonym.

## Replication of Experiment 1: Continuous mouse tracking reveals a 2-stage representation of negated adjectives, in the absence of feedback

We replicated Experiment 1 in a new group of online participants ($N = 55$; **Fig 3**). The experimental procedure was the same as that of Experiment 1, except that no feedback was provided to participants based on the final interpretation, but only if the cursor's movement violated the warnings provided during the familiarization phase (e.g., "you crossed the vertical borders"; see **Materials and methods**). We performed the same data analyses performed for Experiment 1.

**Reaction times.** The 2 (*antonym*: low versus high) × 2 (*negation*: negated versus affirmative) repeated-measures ANOVA reveal a significant main effect of antonyms (F(1,54) = 36.90, $p < 0.001$, $\eta_p^2 = 0.40$) and a significant main effect of negation ($F(1,54) = 73.04$, $p < 0.001$, $\eta_p^2 = 0.57$). Moreover, a significant crossover interaction between antonyms and negation was found (F(1,54) = 16.40, $p < 0.001$, $\eta_p^2 = 0.23$, **Fig 3A**). These results replicate Experiment 1, showing that participants were faster for high adjectives (e.g., "good") than for low adjectives (e.g., "bad") and for affirmative phrases (e.g., "really really good") than for negated phrases (e.g., "really not good"). Results on *complexity* reveal that participants were faster for phrases with 2 modifiers, e.g., "not really," than phrases with 1 modifier, e.g., "not ###" ($F(1,54) = 28.87$, $p < 0.001$, $\eta_p^2 = 0.35$, especially in affirmative phrases: complexity by negation interaction $F(1,54) = 6.26$, $p = 0.015$, $\eta_p^2 = 0.10$), again replicating results of Experiment 1 (see Table B in **S3 Table** for pairwise comparisons between each pair of modifiers).

**Continuous mouse trajectories.** The 2 (*antonym*: low versus high) × 2 (*negation*: negated versus affirmative) repeated-measures ANOVA for participants' final interpretations reveal a significant main effect of antonyms (F(1,54) = 166.40, $p < 0.001$, $\eta_p^2 = 0.75$), a significant main effect of negation (F(1,54) = 48.62, $p < 0.001$, $\eta_p^2 = 0.47$), and a significant interaction between antonyms and negation (F(1,54) = 210.13, $p < 0.001$, $\eta_p^2 = 0.80$). Post hoc tests show that the final interpretation of negated phrases was located at a more central portion of the semantic scale than that of affirmative phrases (affirmative low < negated high, and affirmative high > negated low, $p_{holm} < 0.001$, **Fig 3B**), indicating that negation never inverts the interpretation of adjectives to that of their antonyms. Results also show that the final interpretations of negated phrases was significantly more variable (measured as standard deviations) than that of affirmative phrases (F(1,54) = 15.43, $p < 0.001$, $\eta_p^2 = 0.22$). These results again replicate Experiment 1. As for Experiment 1, we then performed regression analyses with antonyms as the

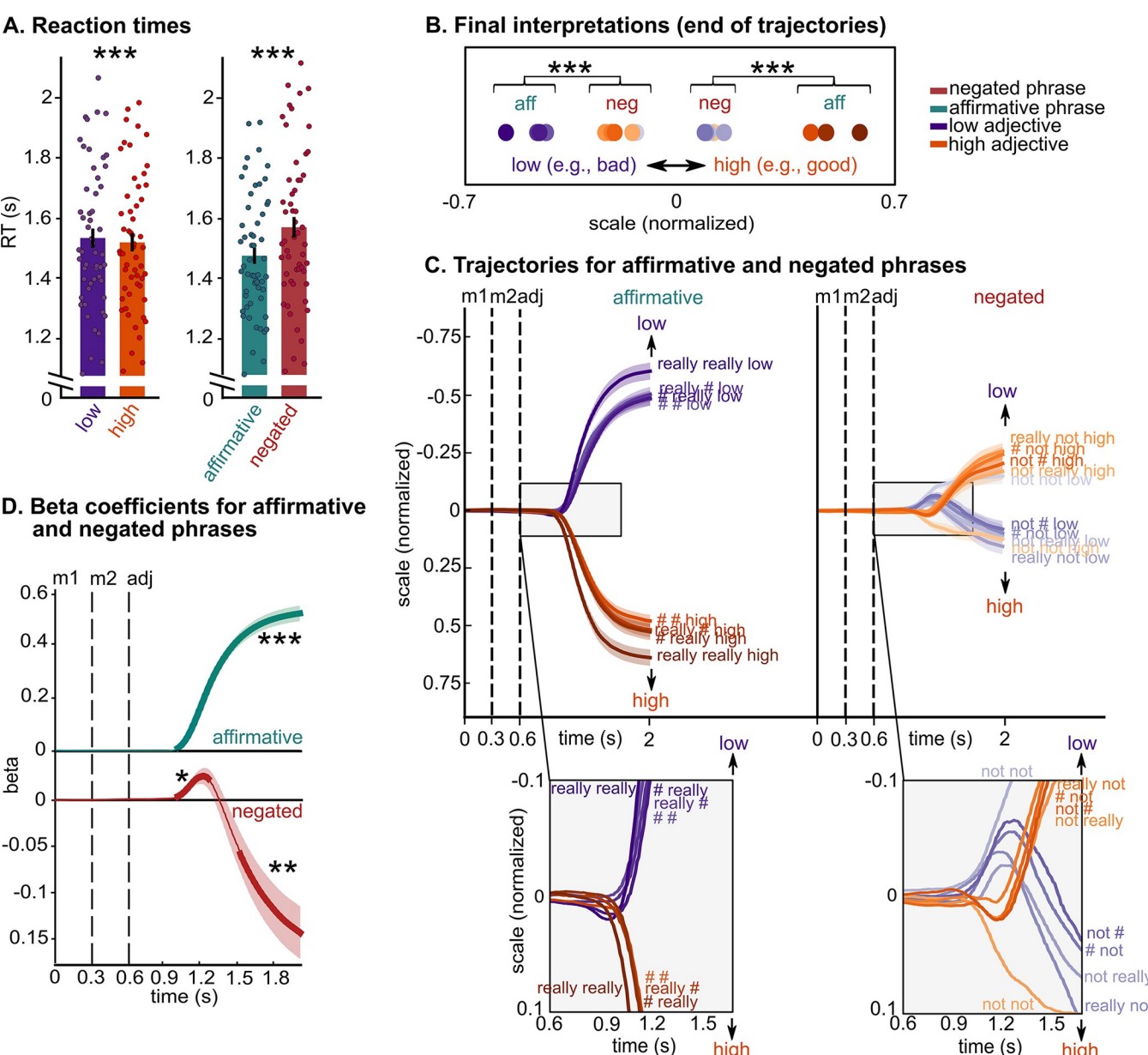

**Fig 3. Replication of Experiment 1, without feedback on interpretation. (A)** Reaction times results for the online behavioral study (*N* = 55). Bars represent the participants' mean ± SEM, and dots represent individual participants. Participants were faster for high adjectives (e.g., "good") than for low adjectives (e.g., "bad") and for affirmative phrases (e.g., "really really good") than for negated phrases (e.g., "really not good"). These results replicate Experiment 1. **(B)** Final interpretations (i.e., end of trajectories) of each phrase, represented by filled circles (purple = low, orange = high), averaged across adjective dimensions and participants, showing that negation never inverts the interpretation of adjectives to that of their antonyms. **(C)** Mouse trajectories for low (purple) and high (orange) antonyms, for each modifier (shades of orange and purple) and for affirmative (left panel) and negated (right panel) phrases. Zoomed-in panels at the bottom demonstrate that mouse trajectories of affirmative phrases branch towards the adjective's side of the scale and remain on that side until the final interpretation; in contrast, the trajectories of negated phrases first deviate towards the side of the adjective and subsequently towards the side of the antonym (except for "not not"). This result is confirmed by linear models fitted to the data at each time point in **D**. These results also replicate Experiment 1. **(D)** Beta values (average over 55 participants) over time, separately for affirmative and negated phrases. Thicker lines indicate significant time windows. Trials with "not not" were not included in this analysis as the trajectories pattern was different compared to the other conditions with negation. **(C, D)** Black vertical dashed lines indicate the presentation onset of each word: modifier 1, modifier 2 and adjective; each line and shading represent participants' mean ± SEM. **(A, B, D)** *** *p* < 0.001; ** *p* < 0.01; * *p* < 0.05. Data are available on the Open Science Framework https://doi.org/10.17605/OSF.IO/5YS6B.

predictor and mouse trajectories as the dependent variable. For this analysis, trials with "not not" were not included as, in this experiment, the trajectories pattern was different compared to the other conditions with negation (**Fig 3C**). The results of the regression analyses show that (1) in affirmative phrases, betas are positive (i.e., mouse trajectories moving towards the adjective) starting from 400 ms from the adjective onset ($p < 0.001$, green line in **Fig 3D**); and that (2) in negated phrases, betas are positive (i.e., mouse trajectories moving towards the adjective) between 400 and 650 ms from the adjective onset ($p = 0.02$), and only became negative (i.e., mouse trajectories moving towards the antonym) from 910 ms from the adjective onset ($p = 0.003$, i.e., red line in **Fig 3D**). This pattern replicates that of Experiment 1.

The replication of Experiment 1 illustrates the robustness of the behavioral mouse tracking findings, even in the absence of feedback. Taken together, these results suggest that participants initially interpreted negated phrases as affirmative (e.g., "not good" interpreted along the "good" side of the scale) and later as a mitigated interpretation of the opposite meaning (e.g., the antonym "bad").

## Experiment 2: MEG shows that negation weakens the representation of adjectives and recruits response inhibition networks

In this study (MEG experiment, $N = 26$), participants read adjective phrases comprising 1 or 2 modifiers ("not" and "really") and scalar adjectives across different dimensions (e.g., "really really good," "really not quiet," "not ### dark"). Adjectives were selected to represent opposite poles (i.e., the antonyms) of the respective semantic scales: *low* pole of the scale (e.g., "bad," "cool," "quiet," "dark") and *high* pole of the scale (e.g., "good," "warm," "loud," "bright"). A sequence of dashes was used to indicate the absence of a modifier. **Fig 1B** and **S2 Table** provide the comprehensive list of the linguistic stimuli. Participants were asked to indicate whether a probe (e.g., 6) represented the meaning of the phrase on a scale from "really really low" (0) to "really really high" (8) (*yes/no* answer, **Fig 1B**). Feedback consisted of a green or red cross, to which 1 and 0 was assigned to compute the average feedback score. Behavioral data of Experiment 2 replicate that of Experiment 1: Negated phrases are processed slower and with lower feedback score than affirmative phrases (main effect of negation for RTs: $F(1,25) = 26.44$, $p < 0.001$, $\eta_p^2 = 0.51$; main effect of negation for feedback score: $F(1,25) = 8.03$, $p = 0.009$, $\eta_p^2 = 0.24$).

The MEG analyses, using largely temporal and spatial decoding approaches [48], comprise 4 incremental steps: (1) we first identify the temporal correlates of simple word representation (i.e., the words "really" and "not" in the modifier position, and each pair of scalar adjectives in the second word position, i.e., the head position; see **S2 Table**); (2) we test lexical-semantic representations of adjectives over time beyond the single word level, by entering *low* ("bad," "cool," "quiet," and "dark") and *high* ("good," "warm," "loud," and "bright") antonyms in the same model (adjectives in purple versus orange in **S2 Table**). We then test the representation of the negation operator over time (modifiers in green versus red in **S2 Table**); (3) we then ask how negation operates on the representation of adjectives, by teasing apart 4 possible mechanisms (i.e., *No effect, Mitigation, Inversion, Change*; adjectives in purple versus orange for modifiers in green and red separately in **S2 Table**); (4) we explore changes in beta power as a function of negation (motivated by the literature implicating beta-band neural activity in linguistic processing).

### (1) *Temporal decoding of single word processing*

The butterfly (bottom) and topography plots (top) in **Fig 4A** illustrate the grand average of the event-related fields elicited by the presentation of all words, as well as the probe, regardless

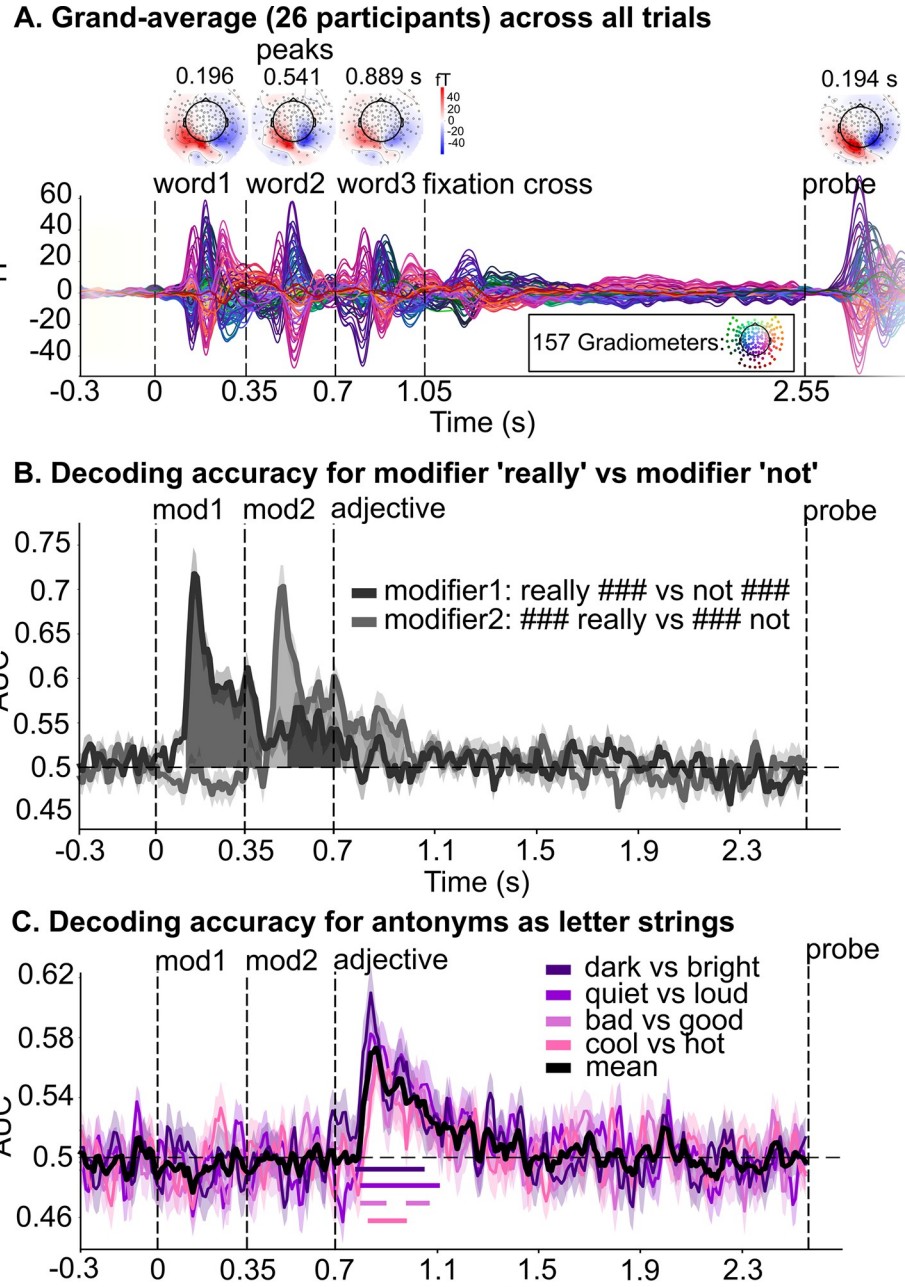

**Fig 4. Evoked activity and temporal decoding of modifiers and adjectives as letter strings. (A)** The butterfly (bottom) and topo plots (top) illustrate the event-related fields elicited by the presentation of each word as well as the probe, with a primarily visual distribution of neural activity right after visual onset (i.e., letter string processing). We performed multivariate decoding analyses on these preprocessed MEG data, after performing linear dimensionality reduction (see **Materials and methods**). Detector distribution of MEG system in inset box. fT: femtoTesla magnetic field strength. **(B)** We estimated the ability of the decoder to discriminate "really" vs. "not" separately in the first and second modifier's position, from all MEG sensors. We contrasted phrases with modifiers "really ###" and "not ###," and phrases with modifiers "### not" and "### really." **(C)** We evaluated whether the brain encodes representational differences between each pair of antonyms (e.g., "bad" vs. "good"), in each of the 4 dimensions (quality, temperature, loudness, and brightness). The mean across adjective pairs is represented as a solid black line; significant windows are indicated by horizontal solid lines below. **(B** and **C)** AUC = area under the receiver operating characteristic curve, chance = 0.5 (black horizontal dashed line); for all panels: black vertical dashed lines indicate the presentation onset of each word: modifier 1, modifier 2, and adjective; each line and shading represent participants' mean ± SEM. Data are available on the Open Science Framework https://doi.org/10.17605/OSF.IO/5YS6B.

of condition. Results of decoding analyses performed on these preprocessed MEG data (after performing linear dimensionality reduction; see **Materials and methods**) show that the temporal decoding of "really" versus "not" is significant between 120 and 430 ms and between 520 and 740 ms from the onset of the first modifier (dark gray shading, $p < 0.001$ and $p = 0.001$) and between 90 and 640 ms from the onset of the second modifier (light gray shading, $p < 0.001$, **Fig 4B**). Pairs of antonyms from different scales (regardless of specific modifier) were similarly decodable between 90 and 410 ms from adjective onset (quality: 110 to 200 ms, $p = 0.002$ and 290 to 370 ms, $p = 0.018$; temperature: 140 to 280 ms, $p < 0.001$; loudness: 110 to 410 ms, $p < 0.001$; brightness: 90 to 350 ms, $p < 0.001$, **Fig 4C**), reflecting time windows during which the brain represents visual, lexical, and semantic information (e.g., [7,49]). These results further show that single words can be decoded with relatively high accuracy (approximately 70%).

(2) *Temporal and spatial decoding of adjectives and negation*

After establishing that single words' features can be successfully decoded in sensible time windows (see **Fig 4**), we moved beyond single word representation and clarified the temporal patterns of adjective and negation representation independently from their interaction and identified temporal windows where to expect changes in adjective representation as a function of negation. First, we selectively evaluated lexical-semantic differences between *low* ("bad," "cool," "quiet," and "dark") and *high* ("good," "warm," "loud," and "bright") adjectives, regardless of the specific scale (i.e., pooling over *quality*, *temperature*, *loudness*, and *brightness*) and by pooling over all modifiers. Temporal decoding analyses (see **Materials and methods**) reveal significant decodability of *low* versus *high* antonyms in 3 time windows between 140 and 560 ms from adjective onset (140 to 280 ms, $p < 0.001$; 370 to 460 ms: $p = 0.009$; 500 to 560 ms: $p = 0.044$, purple shading in **Fig 5A**). No significant differences in lexical-semantic representation between *low* and *high* antonyms were observed in later time windows (i.e., after 560 ms from adjective onset). The spatial decoding analysis illustrated in **Fig 5B** (limited to 50 to 650 ms from adjective onset; see **Materials and methods**) show that decoding accuracy for *low* versus *high* antonyms is significantly above chance in a widespread left-lateralized brain network, encompassing the anterior portion of the superior temporal lobe, the middle, and the inferior temporal lobe (purple shading in **Fig 5B**, significant clusters are indicated by a black contour: left temporal lobe cluster, $p = 0.002$). A significant cluster was also found in the right temporal pole, into the insula ($p = 0.007$). Moreover, we found significant clusters in the bilateral cingulate gyri (posterior and isthmus) and precunei (left precuneus/cingulate cluster, $p = 0.009$; right precuneus/cingulate cluster, $p = 0.037$). Overall, these regions are part of the (predominantly left-lateralized) frontotemporal brain network that underpins lexical-semantic representation and composition [7,8,46,49–55].

Next, we turn to representations of negation over time. We performed a temporal decoding analysis for phrases containing "not" versus phrases not containing "not," separately for phrases with 1 and 2 modifiers (to account for phrase complexity; see **S2 Table** for a list of all trials). For phrases with 1 modifier, the decoding of negation is significantly higher than chance throughout word 1 (−580 to −500 ms from adjective onset, $p = 0.005$), then again throughout word 2 (−470 to 0 ms from adjective onset, $p < 0.001$). After the presentation of the adjective, negation decodability is again significantly above chance between 0 and 40 ms ($p = 0.034$) and between 230 and 290 ms from adjective onset ($p = 0.018$; dark red line and shading in **Fig 5C**). Similarly, for phrases with 2 modifiers, the decoding of negation is significantly higher than chance throughout word 1 (−580 to −410 ms from adjective onset, $p = 0.002$), throughout word 2 (−400 to 0 ms from adjective onset, $p < 0.001$), and for a longer time window from adjective onset compared to phrases with one modifier, i.e., between 0 and

## A. Temporal decoding of antonyms: word meaning

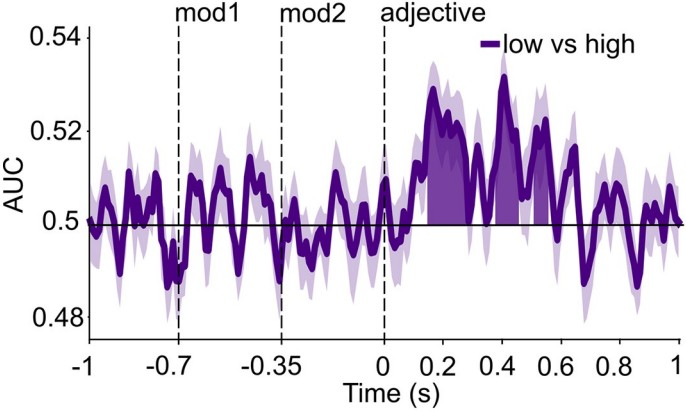

## B. Spatial decoding of antonyms

50–650 ms from the onset of the adjective

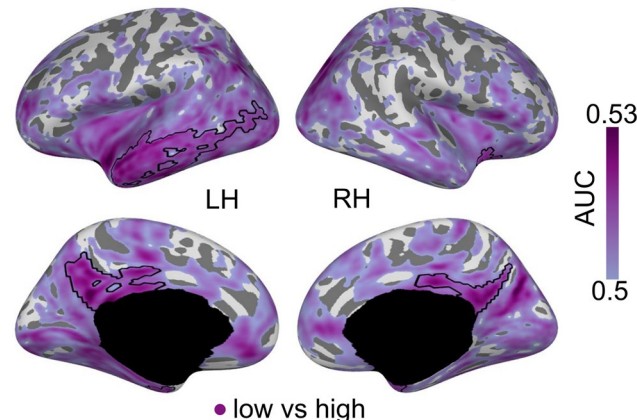

## C. Temporal decoding of negation as a function of complexity

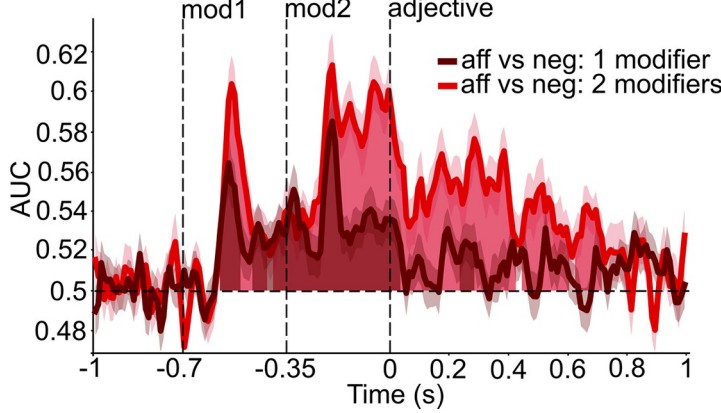

**Fig 5. Temporal and spatial decoding of antonyms across all scales and temporal decoding of negation.** (**A**) Decoding accuracy (purple line) of lexical-semantic differences between antonyms across all scales (i.e., pooling over "bad," "cool," "quiet," and "dark"; and "good," "warm," "loud," and "bright" before fitting the estimators) over time, regardless of modifier; significant time windows are indicated by purple shading. (**B**) Decoding accuracy (shades of purple) for antonyms across all scales over brain sources (after pooling over the 4 dimensions), between 50 and 650 ms

from adjective onset. Significant spatial clusters are indicated by a black contour. (**C**) Decoding accuracy of negation over time, as a function of the number of modifiers (1 modifier: dark red line and shading; 2 modifiers: light red line and shading). 1 modifier: "really ###," "### really," "not ###," "### not"; 2 modifiers: "really really," "really not," "not really," "not not." Significant time windows are indicated by dark red (1 modifier) and light red (2 modifiers) shading. For all panels: AUC: area under the receiver operating characteristic curve, chance = 0.5 (black horizontal dashed line); black vertical dashed lines indicate the presentation onset of each word: modifier1, modifier2, and adjective; each line and shading represent participants' mean ± SEM; aff = affirmative, neg = negated; LH = left hemisphere; RH = right hemisphere. Data are available on the Open Science Framework https://doi.org/10.17605/OSF.IO/5YS6B.

720 ms (0 to 430 ms, $p < 0.001$; 440 to 500 ms, $p = 0.030$; 500 to 610 ms, $p < 0.001$; 620 to 720 ms, $p < 0.001$; light red line and shading in **Fig 5C**). The same analysis time-locked to the onset of the probe shows that negation is once again significantly decodable between 230 and 930 ms after the probe, likely being reinstated when participants perform the task (**S2 Fig**).

Cumulatively, these results suggest that the brain encodes negation every time a "not" is presented and maintains this information up to 720 ms after adjective onset. Further, they show that the duration of negation maintenance is amplified by the presence of a second modifier, highlighting combinatoric effects [2,6,56].

### (3) Effect of negation on lexical-semantic representations of antonyms over time

The temporal decoding analyses performed separately for adjectives and for negation demonstrate that the brain maintains the representation of the modifiers available throughout the presentation of the adjective. Here we ask how negation *operates on* the representation of the antonyms at the neural level, leveraging theoretical accounts of negation [11,12,42–44], behavioral results of Experiment 1, and 2 complementary decoding approaches. We test 4 hypotheses (see *Predictions* in **Fig 6A**): (1) *No effect of negation*: Negation does not change the representation of adjectives (i.e., "not low" = "low"). We included this hypothesis based on the 2-step theory of negation, wherein the initial representation of negated adjectives would not be affected by negation [27]. (2) *Mitigation*: Negation weakens the representation of adjectives (i.e., "not low" < "low"). (3) *Inversion*: Negation inverts the representation of adjectives (i.e., "not low" = "high"). Hypotheses (2) and (3) are derived from previous linguistics and psycholinguistics accounts on comprehension of negated adjectives [42–44]. Finally, (4) *Change*: We evaluated the possibility that negation might change the representation of adjectives to another representation outside the semantic scale defined by the 2 antonyms (e.g., "not low" = e.g., "fair"). Importantly, these predictions focus on *how* negation affects representations rather than on *when*. Thus, a combination of mechanisms may be observed over time (e.g., first *no effect* and then *inversion*).

To adjudicate between these 4 hypotheses, we performed 2 complementary sets of decoding analyses. Decoding approach (i): we computed the accuracy with which estimators trained on *low* versus *high* antonyms in affirmative phrases (e.g., "really really bad" versus "really really good") generalize to the representation of *low* versus *high* antonyms in negated phrases (e.g., "really not bad" versus "really not good") at each time sample time-locked to adjective onset (see **Materials and methods**); decoding approach (ii): we trained estimators on *low* versus *high* antonyms in affirmative and negated phrases together (in 90% of the trials) and computed the accuracy of the model in predicting the representation of *low* versus *high* antonyms in affirmative and negated phrases separately (in the remaining 10% of the trials; see **Materials and methods**). Decoding approach (ii) allows for a direct comparison between AUC and probability estimates in affirmative and negated phrases and to disentangle predictions (1) *No effect* from (2) *Mitigation*. Expected probability estimates (i.e., the averaged class probabilities for *low* and *high* classes) as a result of decoding approach (i) and (ii) are depicted as light and dark, green and red bars under *Decoding approach* in **Fig 6A**.

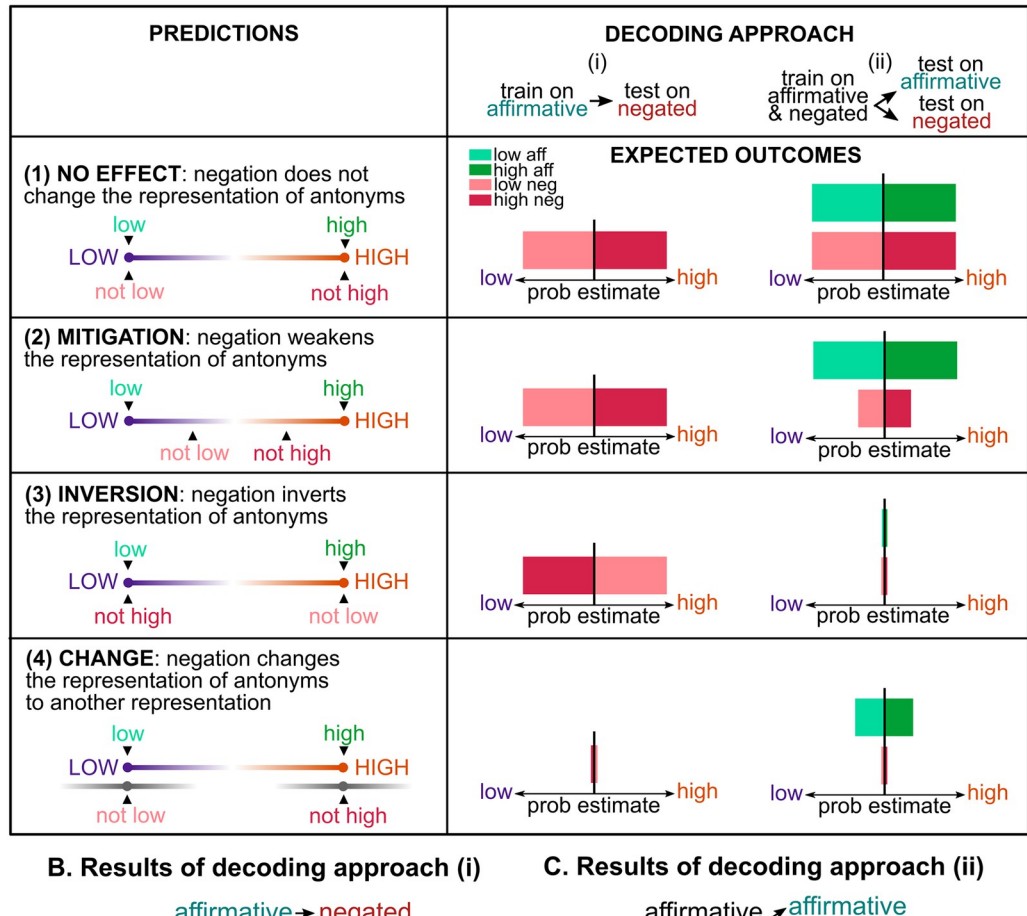

**Fig 6. Predictions, decoding approaches, and results of the effect of negation on the representation of adjectives. (A)**
We tested 4 possible effects of negation on the representation of adjectives: (1) *No effect*; (2) *Mitigation*; (3) *Inversion*; (4)
*Change* (left column). Note that we depicted predictions of (3) *Inversion* on the extremes of the scale, but a combination of
inversion and mitigation would have the same expected outcomes. We performed 2 sets of decoding analyses (right
column): (i) We trained estimators on low (purple) vs. high (orange) antonyms in affirmative phrases and predicted model

accuracy and probability estimates of low vs. high antonyms in negated phrases (light and dark red bars). (ii) We trained estimators on low vs. high antonyms in affirmative and negated phrases together and predicted model accuracy and probability estimates in affirmative (light and dark green bars) and negated phrases (light and dark red bars) separately. (**B**) Decoding accuracy (red line) over time of antonyms for negated phrases, as a result of decoding approach (i). Significant time windows are indicated by red shading and horizontal solid lines. (**C**) Decoding accuracy of antonyms over time for affirmative (green line) and negated (red line) phrases, as a result of decoding approach (ii). Significant time windows for affirmative and negated phrases are indicated by green and red shading and horizontal solid lines. The significant time window of the difference between affirmative and negated phrases is indicated by a black horizontal solid line. (**D**) Probability estimates for low (light red) and high (dark red) negated antonyms averaged across the significant time windows depicted in **B**. Bars represent the participants' mean ± SEM and dots represent individual participants. (**E**) Probability estimates for low (light green) and high (dark green) affirmative adjectives and for low (light red) and high (dark red) negated adjectives, averaged across the significant time window depicted as a black horizontal line in **C**. Chance level of probability estimates was computed by averaging probability estimates of the respective baseline (note that the baseline differs from 0.5 due to the different number of trials for each class in the training set of decoding approach (i)). Bars represent the participants' mean ± SEM and dots represent individual participants. (**B** and **C**) AUC: area under the receiver operating characteristic curve, chance = 0.5 (black horizontal dashed line); each line and shading represent participants' mean ± SEM. (**B–E**) The black vertical dashed line indicates the presentation onset of the adjective; green = affirmative phrases, red = negated phrases. Data are available on the Open Science Framework https://doi.org/10. 17605/OSF.IO/5YS6B.

Temporal decoding approach (i) reveals that the estimators trained on the representation of *low* versus *high* antonyms in affirmative phrases significantly generalize to the representation of *low* versus *high* antonyms in negated phrases, in 4 time windows between 130 and 550 ms from adjective onset (130 to 190 ms, $p = 0.039$; 200 to 270 ms: $p = 0.003$; 380 to 500 ms: $p < 0.001$; 500 to 550 ms: $p = 0.008$; red shading in **Fig 6B**). **Fig 6D** depicts the probability estimates averaged over the significant time windows for *low* and *high* antonyms in negated phrases. These results only support predictions (1) *No effect* and (2) *Mitigation*, thus invalidating predictions (3) *Inversion* and (4) *Change*. **S3 Fig** illustrates a different approach that similarly leads to the exclusion of prediction *(3) Inversion*.

Temporal decoding approach (ii) shows significant above chance decoding accuracy for affirmative phrases between 130 and 280 ms ($p < 0.001$) and between 370 and 420 ms ($p = 0.035$) from adjective onset. Conversely, decoding accuracy for negated phrases is significantly above chance only between 380 and 450 ms after the onset of the adjective ($p = 0.004$). Strikingly, negated phrases are associated with significantly lower decoding accuracy than affirmative phrases in the time window between 130 and 190 ms from adjective onset ($p = 0.040$; black horizontal line in **Fig 6C**). **Fig 6E** represents the probability estimates averaged over this 130 to 190 ms significant time window for *low* and *high* antonyms, separately in affirmative and negated phrases, illustrating reduced probability estimates for negated compared to affirmative phrases. No significant difference between decoding accuracy of affirmative and negative phrases was found for later time windows (500 to 1,000 ms from adjective onset, $p > 0.05$). A follow-up analysis where we trained and tested on *low* versus *high* antonyms in affirmative and negated phrases separately shows similar results (**S4A Fig**). Furthermore, the analysis including all trials, regardless of feedback score, also shows similar results (**S4B Fig**).

Overall, the generalization of representation from affirmative to negated phrases and the higher decoding accuracy (and probability estimates) for affirmative than negated phrases within the first 500 ms from adjective onset (i.e., within the time window of lexical-semantic processing shown in **Fig 5A**) provide direct evidence in support of prediction (2) *Mitigation*, wherein negation weakens the representation of adjectives. The alternative hypotheses did not survive the different decoding approaches.

(4) *Changes in beta power as a function of negation*

We distinguished among 4 possible mechanisms of how negation could operate on the representation of adjectives and demonstrated that negation does not invert or change the

representation of adjectives but rather weakens the decodability of *low* versus *high* antonyms within the first approximately 300 ms from adjective onset (**Fig 6C**; with AUC for affirmative and negated adjectives being significantly different for about 60 ms within this time window). The availability of negation upon the processing of the adjective (**Fig 5A and 5C**) and the reduced decoding accuracy for antonyms in negated phrases (**Fig 6C**) raise the question of whether negation operates through inhibitory mechanisms, as suggested by previous research employing action-related verbal material [35–37]. We therefore performed time-frequency analyses, focusing on beta power (including low-beta: 12 to 20 Hz, and high-beta: 20 to 30 Hz [57]; see **Materials and methods**), which has been previously associated with inhibitory control [58] (see **S5 Fig** for comprehensive time-frequency results). We reasoned that, if negation operates through general-purpose inhibitory systems, we should observe higher beta power for negated than affirmative phrases in sensorimotor brain regions.

Our results are consistent with this hypothesis, showing significantly higher low-beta power (from 229 to 350 ms from the onset of modifier1: $p = 0.036$; from 326 to 690 ms from adjective onset: $p = 0.012$; red line in **Fig 7A**) and high-beta power (from 98 to 271 ms from adjective onset: $p = 0.044$; yellow line in **Fig 7A**) for negated than affirmative phrases. **S6 Fig** further shows low- and high-beta power separately for negated and affirmative phrases, compared to phrases with no modifier (i.e., with "### ###").

Our whole-brain source localization analysis shows significantly higher low-beta power for negated than affirmative phrases in the left precentral, postcentral, and paracentral gyri ($p = 0.012$; between 326 and 690 ms from adjective onset, red cluster in **Fig 7C**). For high-beta power, similar (albeit not significant) sensorimotor spatial patterns emerge (yellow cluster in **Fig 7B**).

## Discussion

We tracked changes over time in lexical-semantic representations of scalar adjectives, as a function of the intensifier "really" and the negation operator "not." Neural correlates of negation have typically been investigated in the context of action verbs [29,35–37,40,41,59–63]. Our study employs minimal linguistic contexts to characterize in detail how negation operates on abstract, non-action-related lexical-semantic representations. We leveraged (1) psycholinguistic findings on adjectives that offer a framework wherein meaning is represented on a continuum [42,43]; (2) time-resolved behavioral and neural data; and (3) multivariate analysis methods (decoding), which can discriminate complex lexical-semantic representations from distributed neuronal patterns (e.g., [62]).

The longer RTs and lower feedback score for negated phases shown in Experiment 1 (**Fig 2A**), in the replication experiment (**Fig 3A**), and in Experiment 2, are consistent with data demonstrating that negation incurs increased processing costs [13–18,27,32]. More significantly, mouse trajectories show that participants initially interpreted negated phrases as affirmative (e.g., "not good" is located on the "good" side of the scale, for approximately 130 ms, **Figs 2C** and **3C**), indicating that initial representations of negated scalar adjectives are closer to the representations of the adjectives rather than that of their antonyms. Similarly, participants' final interpretations of negated adjectives (e.g., "not good," "really not good") never overlapped with the final interpretations of the corresponding affirmative antonyms (e.g., "bad," "really bad," "really really bad"; **Figs 2B** and **3B**), highlighting how negation never inverts the meaning of an adjective to that of its antonym, even when participants are making decisions on a binary semantic scale [9,37–40].

Continuous mouse trajectories allowed us to quantify dynamic changes in participants' interpretations. MEG provided a means to directly track neural representations over time. We

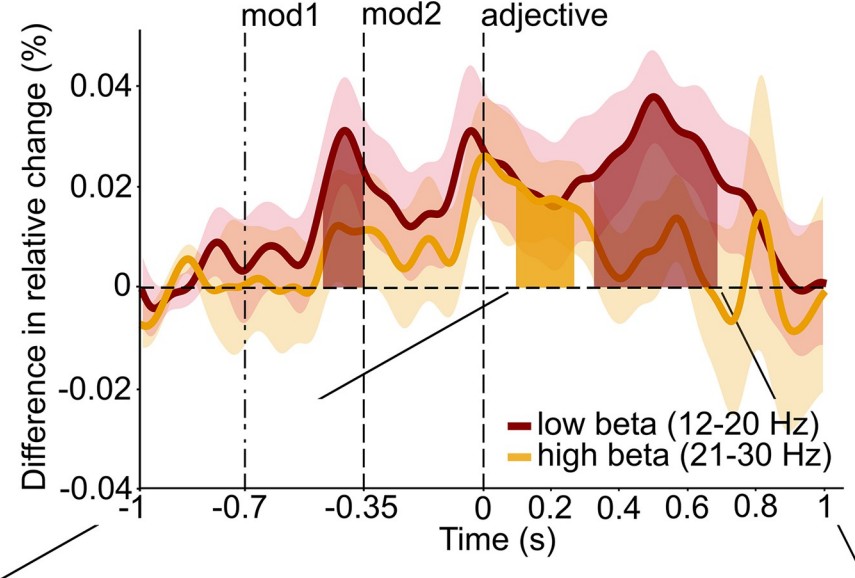

## A. Low- and high-beta power of (negated - affirmative phrases)

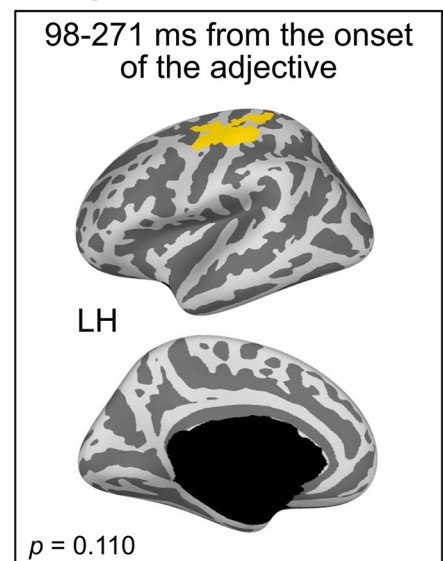

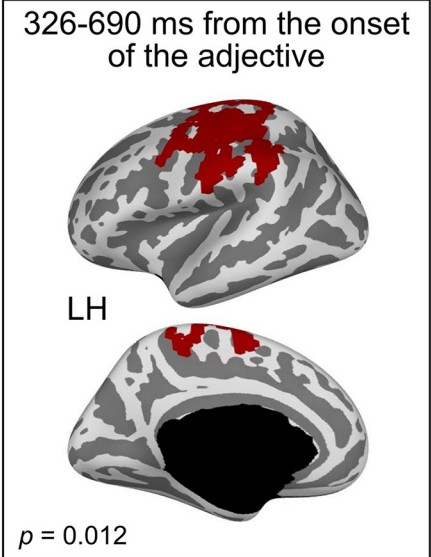

**Fig 7. Differences in beta power over time between negated and affirmative phrases.** (**A**) Differences in low (12–20 Hz, red) and high (21–30 Hz, yellow) beta power over time between negated (i.e., "### not," "not ###," "really not," "not really," "not not") and affirmative phrases (i.e., "### really," "really ###," "really really"). Negated phrases show higher beta power compared to affirmative phrases throughout the presentation of the modifiers and for a sustained time window from adjective onset up to approximately 700 ms; significant time windows are indicated by red (low-beta) and yellow (high-beta) shading; black vertical dashed lines indicate the presentation onset of each word: modifier1, modifier2, and adjective; each line and shading represent participants' mean ± SEM. (**B**) Differences (however not reaching statistical significance, $\alpha = 0.05$) in high-beta power between negated and affirmative phrases (restricted between 97 and 271 ms from adjective onset, yellow cluster). (**C**) Significant differences in low-beta power between negated and affirmative phrases (restricted between 326 and 690 ms from adjective onset) in the left precentral, postcentral, and paracentral gyrus (red cluster). Note that no significant spatial clusters were found in the right hemisphere. Data are available on the Open Science Framework https://doi.org/10.17605/OSF.IO/5YS6B.

first identified the temporal correlates of lexical-semantic processing *separately* for scalar adjectives and for the negation operator. The time window of adjective representation (approximately 140 to 560 ms from adjective onset, **Fig 5A**) is consistent with previous studies investigating lexical-semantic processing in language comprehension (130 to 200 ms up to approximately 550 ms from adjective onset [64–68]). Spatial decoding results corroborate temporal results, highlighting the involvement of the left-lateralized frontotemporal brain network in adjective processing (**Fig 5B**; [7,8,46,49–55]). Our data further show that negation is processed up to approximately 700 ms from adjective onset (**Fig 5C**). Overall, these data demonstrate that both scalar adjectives and negation are represented between 140 and 560 ms from adjective onset (compare **Fig 5A and 5C**), suggesting that they are represented in parallel and not serially (i.e., one after the other; see [69,70] for related patterns in the context of negation + auxiliary verb and adjective + noun). Finally, they show that the decodability of negation increases in phrases with 2 modifiers (e.g., "really not," "not really," **Figs 5C and S2**), highlighting compositional effects [6].

We then evaluated the effects of the negation operator *on* adjective representation, to address the question of *how* negation operates on lexical-semantic representations of antonyms. We contrasted 4 hypotheses (**Fig 6A**): Negation (1) does not change the representation of scalar adjectives (e.g., "not good" = "good," *No effect*); (2) weakens the representation of scalar adjectives (e.g., "not good" < "good," *Mitigation*); (3) inverts the representation of scalar adjectives (e.g., "not good" = "bad," *Inversion*); or (4) changes the representation of scalar adjectives to another representation (e.g., "not good" = e.g., "unacceptable," *Change*). These 4 hypotheses make predictions about how negation operates on scalar adjectives at any given time. It is thus possible that multiple mechanisms may unfold over time when looking at time-resolved data (e.g., first *no effect* and then *inversion*). Using 2 complementary decoding approaches, we demonstrated that, within the time window of adjective encoding, the representation of affirmative adjectives generalizes to that of negated adjectives (**Fig 6B and 6D**). This finding rules out predictions (3) *Inversion* and (4) *Change*. Moreover, these findings complement our behavioral data that show that negated adjectives are initially interpreted by participants as affirmative. Second, we showed that the representation of adjectives in affirmative and negated phrases is not identical but is weakened by negation (**Fig 6C and 6E**). This result rules out prediction (1) *No effect* and supports prediction (2) *Mitigation*, wherein negation weakens the representation of adjectives. We observed such a reduction in early representations (i.e., within approximately 300 ms from adjective onset). This finding is consistent with previous research that reported effects of negation as soon as lexical-semantic representations of words are formed [12,29–31,71], and not exclusively at later processing stages (e.g., P600 [72,73]). In addition, the fact that *low* versus *high* adjectives are decodable approximately 400 ms after the adjective onset in negated phrases (**Figs 6C, S4A, and S4B**) raises 2 novel questions: First, is the mitigation effect of negation stable over time? Second, at what exact stages does it operate upon (see [64,66])? Using a masked priming paradigm, van Gaal and colleagues [33] analyzed participants' EEG responses to sequences of words that were either consciously or unconsciously perceived. Their findings indicate that the meaning of multiple words, including negation, can be integrated even when participants report not seeing them, but that conscious perception is required for later grammatical integration. Future research remains necessary to more precisely tease apart the lexical, semantic, and syntactic features that are selectively affected by the negation operator over time.

Taken together, our behavioral and neural data jointly point to a *mitigation* rather than an *inversion* effect of negation at early semantic processing stages and exclude the hypothesis according to which negation does not change the representation of antonyms (i.e., first processing step in 2-step accounts [27]). Specifically, these results show that initial interpretations

and early neural representations of negated adjectives are similar to that of affirmative adjectives, but weakened. One possible limitation is that visual features might still be deciphered and consequently affect the analyses presented in **Fig 6**. If the consistency or probability of visual features influences the decoding performance, discriminating between the predicted patterns for our competing hypotheses may become more difficult. For example, if the underlying representation is a changed representation (see 4 in **Fig 6**), but part of the decoding performance is attributable to visual consistencies, that could effectively increase the overall decoding performance, leading to a pattern consistent with mitigation (see 2 in **Fig 6**). Two factors mitigate this potential issue: First, our stimulus set by design contains words with varying visual features, which makes it difficult for a linear decoder to identify a common visual feature to rely on; Second, generalized representations shown for decoding approach (i) last up to 550 ms from adjective onset, likely reflecting lexical-semantic rather than purely perceptual processing [64–68]. The comparison between MEG and behavioral results also reveals interesting differences. Behavioral data reveal that, in negated phrases, participants later modify their initial interpretation towards, but never exactly as, the opposite meaning. Our MEG data do not show an inversion of adjective representation as a function of negation, at early or later lexical-semantic processing stages. Differences between our behavioral and neural results could be ascribed to the fact that the behavioral task had to be adapted to the MEG environment. In the behavioral experiment (and its replication), participants were continuously and explicitly indicating their interpretations in a bidimensional space (i.e., the semantic scale defined by the polar adjectives). Conversely, the MEG experiment allowed us to measure neural correlates of semantic representation over time (not affected by motor activity) but not the explicit interpretations, which were collected only after the probe was presented (1,850 ms after the adjective presentation) and only through a yes/no decision task (**Figs 2B** and **4A**). Thus, both our designs present advantages and limitations and together offer a more encompassing approach: Bidimensional semantic scales in the behavioral task could have masked nuanced changes in semantic representation over time, which we were able to better evaluate in the MEG data; in turn, later effects of negation might have been masked by the task employed in the MEG experiment; however, they were more carefully assessed in our behavioral approach. Design limitations are crucial when interpreting behavioral and neuroimaging findings. Adopting a multimodality approach (e.g., behavioral mouse tracking and MEG recordings) helps overcome some of the limitations related to individual experimental approaches; our complementary approaches allowed us to advance a well-motivated interpretation.

While previous fMRI studies on sentential negation have shown that negation reduces hemodynamic brain activations related to verb processing [40,41], the current study offers novel time-resolved behavioral and neural data on how negation selectively operates on abstract concepts. Previous research has highlighted that negation might behave differently depending on the pragmatics of discourse interpretation, e.g., when presented in isolation as compared to when presented in context ("not wrong" versus "this theory is not wrong" [9,10]), or when used ironically ("they are not really good" said ironically to mean that they are "mediocre," e.g., [11,71]). Within this pragmatic framework, it has been suggested that the opposite meaning of a scalar adjective would be more simply conveyed by the affirmative counterpart than by negation [11,44,74]; thus, to convey the opposite meaning of "bad," it would be more appropriate to use "good" as opposed to "not bad." Following this logic, negation would be purposefully used (and understood) to convey a different, mitigated meaning of the adjective (e.g., "not bad" = "less than bad"). Although we did not directly manipulate sentential or pragmatic contexts, our findings provide behavioral and neural evidence that negation acts as a mitigator. Here, we only tested adjective pairs that form *contraries* (which lie on a continuum, e.g., "bad" and "good"); thus, inherently different patterns of results could emerge

in the case of *contradictories* (which form a dichotomy, e.g., "dead" and "alive"; [44]), where there is no continuum for mitigation to have an effect.

Overall, evidence that negation weakens adjective representations invites the hypothesis that negation operates as a suppression mechanism, possibly through general-purpose inhibitory systems [36,37]. To address this, we compared beta power modulations in affirmative and negated phrases (**Fig 7**). In addition to subserving motor processing, beta-power modulation (12 to 30 Hz) has been associated with attention and expectancy violation and with multiple aspects of language processing, such as semantic memory and syntactic binding, as well as feedback processing ([35,75–78]; for a review, see [57,79]). We evaluated differences between negated and affirmative phrases separately in the low- and high-beta bands. We found greater power for negated than affirmative phrases in both bands, during the processing of the modifier and throughout the processing of the adjective up to approximately 700 ms, localized in left-lateralized sensorimotor areas. The timing and spatial correlates of beta-power in relation to negation align with studies that examined the effect of negation on (mental and motor) action representation [36]. Strikingly, we demonstrated that negation recruits brain areas and neurophysiological mechanisms similar to that recruited by response inhibition—however, in the absence of action-related language material. Within a framework that recognizes 2 interactive neural systems, i.e., a semantic representation and a semantic control system [53], negation would operate through the latter, modulating how activation propagates through the (ventral) language semantic network wherein meaning is represented. The precise connectivity that underpins mitigation of lexical-semantic representations remains to be investigated.

It is uncontroversial that negation is a fundamental operation in language and cognition. A comprehensive cognitive model of negation has been elusive. Our design (**Fig 1**), the explicit hypothesis space we consider (**Fig 6A**), and the choice of methodologies we employ derive from the careful consideration of the different theoretical frameworks [11,12,27,28,44]. To be sure, we also built on existing behavioral measures (i.e., response time and accuracies; [17]), psycholinguistic approaches (i.e., effects of negation on scalar adjectives; [42,43]), and neuroimaging domains (i.e., time, space, and frequency domains; [26,35,40]) that have been used in the study of negation. Our insights contribute decisive new data to the debate on how negation operates by showing that negation functions as a mitigator of lexical-semantic representations of scalar adjectives, possibly through general-purpose inhibitory systems. The results also speak to the question of when negation operates by showing early effects of negation on adjective processing. More broadly, we show that, by characterizing subtle changes of linguistic meaning through negation, using time-resolved behavioral and neuroimaging methods and multivariate decoding, we can tease apart different possible representational outcomes of combinatorial operations, above and beyond the sum of the processing of individual word meanings.

## Materials and methods

### Participants

**Experiment 1 (and replication): Continuous behavioral tracking.**  A total of 101 participants (46 females; mean age = 29.6 years; range 18 to 67 years) completed an online mouse tracking experiment. Participants were recruited via Amazon Mechanical Turk and via the platform SONA (a platform for students' recruitment). All participants were native English speakers with self-reported normal hearing, normal or corrected to normal vision, and no neurological deficits. A total of 97 participants were right-handed. Participants were paid or granted university credits for taking part in the study, which was performed online. All participants provided written informed consent, as approved by the local institutional review board

(New York University's Committee on Activities Involving Human Subjects). The experimental protocol was conducted in accordance with the Declaration of Helsinki. The data of 23 participants were excluded from the data analysis due to (i) number of "incorrect" feedback (based on the warnings) >30%; (ii) mean RTs > 2 SD from the group mean; or (iii) response trajectory always ending within 1/4 from the center of the scale, regardless of condition (i.e., participants who did not pay attention to the instructions of the task). Thus, 78 participants were included in the analyses. The sample size was determined based on previous studies using a similar behavioral approach (approximately 30 participants [15,45,80]) and was increased to account for the exclusion rate reported for online crowdsourcing experiments [81,82].

A new group of 60 participants (37 females; mean age = 19.26 years; range 18 to 23 years) completed the online mouse tracking replication experiment. Participants were recruited via the platform SONA. All participants were native English speakers with self-reported normal hearing and no neurological deficits. A total of 59 participants were right-handed. Participants were granted university credits for taking part in the study, which was performed online. All participants provided written informed consent, as approved by the local institutional review board (New York University's Committee on Activities Involving Human Subjects). The data of 5 participants were excluded from the data analysis due to (i) number of "incorrect" feedback based on the warnings >30%; (ii) mean RTs > 2 SD from the group mean; or (iii) response trajectory always ending within 1/4 from the center of the scale, regardless of condition (i.e., participants who did not pay attention to the instructions of the task). Thus, 55 participants were included in the analyses.

**Experiment 2: MEG.**   A new group of 28 participants (17 females; mean age = 28.7 years; range 19 to 53 years) took part in the in-lab MEG experiment. All participants were native English speakers with self-reported normal hearing, normal or corrected to normal vision, and no neurological deficits. A total of 24 participants were right-handed. They were paid or granted university credits for taking part in the study. All participants provided written informed consent, as approved by the local institutional review board (New York University's Committee on Activities Involving Human Subjects). The experimental protocol was conducted in accordance with the Declaration of Helsinki. The data of 2 participants were excluded from the data analysis because their feedback scores in the behavioral task was <60%. Thus, 26 participants were included in the analysis. The sample size was determined based on previous studies investigating negation using EEG (17 to 33 participants [26,35,37]), investigating semantic representation using MEG (25 to 27 participants [7,8]), or employing decoding methods with MEG data (17 to 20 participants [83,84]).

### Stimuli, design, and procedure

**Experiment 1 (and replication): Continuous mouse tracking.**
**Stimuli and design**. The linguistic stimulus set comprises 108 unique adjective phrases (for the complete list, see **S1 Table**). Adjectives were selected to be antonyms (i.e., *low* and *high* poles of the scale) in the following 6 cognitive or sensory dimensions: *quality* ("bad," "good"), *beauty* ("ugly," "beautiful"), *mood* ("sad," "happy"), *temperature* ("cold," "hot"), *speed* ("slow," "fast"), and *size* ("small," "big"). These antonyms are all *contraries* (i.e., adjectives that lie on a continuum [44]). Lexical characteristics of the antonyms were balanced according to the English Lexicon Project [85]; mean (SD) HAL log frequency of *low* adjectives: 10.69 (1.09), *high* adjectives: 11.51 (1.07), mean (SD) bigram frequency of *low* adjectives: 1,087.10 (374), *high* adjectives: 1,032 (477.2); mean (SD) lexical decision RTs of *low* adjectives: 566 (37), *high* adjectives: 586 ms (70)). Adjectives were combined with 0 (e.g., "### ###"), 1 (e.g., "really

###"), or 2 modifiers (e.g., "really not"). Modifiers were either the intensifier "really" or the negation "not" (see [33] for a similar choice of modifiers; "really" was preferred to "very" as it more strongly intensifies the meaning of the adjective, e.g., "really hot" > "very hot"). A sequence of dashes was used to indicate the absence of a modifier, e.g., "really ### good." Each of the 12 adjectives was preceded by each of the 9 possible combinations of modifiers: "### ###," "### really," "really ###," "### not," "not ###," "really not," "not really," "really really," and "not not," to diversify modifiers' sequences and measure how negation affects adjective representation above and beyond the specific effects of the words "really" and "not." Note that "not not" was included to achieve a full experimental design, even if it is not a frequent combination in natural language and its cognitive and linguistic representations are still under investigation (see [86]). Each dimension (e.g., quality) was presented in 2 blocks (1 block for each scale orientation, e.g., *low* to *high* and *high* to *low*) for a total of 12 blocks. Each phrase was repeated 3 times within each block (note that "### really"/"really ###" were repeated an overall of 3 times, and so were "### not"/"not ###"). Thus, the overall experiment comprised 504 trials. The order of phrases was randomized within each block for each participant. The order of pairs of blocks was randomized across participants.

**Procedure**. Behavioral trajectories provide time-resolved dynamic data that reflect changes in representation [15,45,47]. The online experiment was developed using oTree, a Python-based framework for the development of controlled experiments on online platforms [87]. Participants performed this study remotely, using their own monitor and mouse (touchpads were not allowed). They were instructed to read affirmative or negated adjective phrases (e.g., "really really good," "really not bad") and rate the overall meaning of each phrase on a scale, e.g., from "really really bad" to "really really good." Participants were initially familiarized with the experiment through short videos and a short practice block (18 trials with feedback). They were instructed that the poles of the scale (e.g., "bad" and "good") would be reversed in half of the trials and warned that (i) they could not cross the vertical borders of the response space; (ii) they had to maintain a constant velocity, by following an horizontal line moving vertically; and (iii) they could not rate the meaning of the phrase before the third word was presented. At the beginning of each trial, a response area of 600 (horizontal) × 450 (vertical) pixels and a solid line at the top of the rectangle were presented (**Fig 1A**). Participants were informed about the scale (e.g., quality) and the direction of the scale (e.g., "bad" to "good" or "good" to "bad," i.e., 1 to 10 or 10 to 1). Participants were instructed to click on the "start" button and move the cursor of the mouse to the portion of the scale that best represented the overall meaning of the phrase. The "start" button was placed in the center portion of the bottom of the response space (i.e., in a neutral position). Once "start" was clicked on, information about the scale and scale direction disappeared, leaving only the solid line on screen. Phrases were presented at the top of the response space, from the time when participants clicked on "start," one word at a time, each word for 250 ms (inter-word-interval: 50 ms). After each trial, participants were provided the "incorrect" feedback if the cursor's movement violated the warnings provided during the familiarization phase, and an explanation was provided (e.g., "you crossed the vertical borders"). To keep participants engaged, we provided feedback also based on the final interpretation: "negative" if the response was in the half of the scale opposite to the adjective (for the conditions: "### ###," "#### really," "really ###," and "really really"), or in the same half of the scale of the adjective (for the conditions: "### not" or "not ###"), or in the outer 20% left and right portions of the scale (for the conditions: "really not," "not really," and "not not"); feedback was "positive" otherwise. In case of a trial with negative feedback, the following trial was delayed for 4 seconds. For each trial, we collected continuous mouse trajectories and RTs. The overall duration of the behavioral experiment was approximately 90 minutes. To verify that

the feedback did not affect our results, we ran a replication study with a new group of 55 online participants where no feedback was provided based on the final interpretation.

**Experiment 2: MEG.**

**Stimuli and design.** The linguistic stimulus set comprised 72 unique adjective phrases (for the complete list, see **S2 Table**). Similar to Experiment 1, adjectives were selected for being antonyms (and *contraries*) in the following cognitive or sensory dimensions (touch, audition, vision): *quality* ("bad," "good"), *temperature* ("cool," "warm"), *loudness* ("quiet," "loud"), and *brightness* ("dark," "bright"). The number of semantic scales (4) represents a trade-off between stimulus variability, number of stimuli within each condition—which is essential to achieve a reliable decoding accuracy—and experiment duration for attention maintenance. Lexical characteristics of the antonyms were balanced according to the English Lexicon Project ([85]; mean (SD) HAL log frequency of "low" adjectives: 10.85 (1.03), "high" adjectives: 10.55 (1.88); mean (SD) bigram frequency of "low" adjectives: 1,196.5 (824.6), "high" adjectives: 1,077.5 (376.3); mean (SD) lexical decision RTs of "low" adjectives: 594 ms (39), "high" adjectives: 594 (33)). Adjectives were combined with 0 (e.g., "### ###"), 1 (e.g., "really ###"), or 2 modifiers (e.g., "really not"). Modifiers were either the intensifier "really" or the negation "not." A sequence of dashes was used to indicate the absence of a modifier, e.g., "really ### good." Each of the 8 adjectives was preceded by each of the 9 possible combinations of modifiers: "### ###," "#### really," "really ###," "### not," "not ###," "really not," "not really," "really really," and "not not" ("not not" was included to achieve a full experimental design, even if it is not a frequent combination in natural language. See **S4C, S4D, and S4E Fig** where we speculate that 2 "not," i.e., double negation, do not cancel each other out but rather have mitigation effects similar to that of "really not"). To avoid possible differences in neural representation of phrases with and without syntactic/semantic composition, the condition with no modifiers ("### ###") was exclusively employed as a baseline comparison in the time-frequency analysis and was excluded from all other analyses. Each dimension (e.g., quality) was presented in 2 blocks, 1 block for each yes/no key orientation (8 blocks in total; see Procedure). Each phrase (e.g., "really really bad") was repeated 4 times within 1 block. Thus, the overall experiment comprised 576 trials. The order of phrases was randomized within each block for each participant. The order of blocks was randomized across participants within the first and second half of the experiment. The yes/no order was randomized across participants.

**Procedure.** Participants were familiarized with the linguistic stimuli through a short practice block that mimicked the structure of the experimental blocks. They were instructed to read affirmative or negated adjective phrases (e.g., "really really good," "really not bad") and derive the overall meaning of each adjective phrase, on a scale from 0 to 8, e.g., from "really really bad" to "really really good." Each trial started with a fixation cross (duration: 750 ms), followed by each phrase presented one word at a time, each word for 100 ms (inter-word-interval: 250 ms, **Fig 1B**). After each phrase, a fixation cross was presented for 1,500 ms. A number (i.e., probe) was then presented. To keep the task engaging, participants were required to indicate whether the probe number represented the meaning of the phrase on the scale (*yes/no* answer). The order of the yes/no response keys was swapped halfway through the experiment. Responses had no time limit. If matching (+/− one step on the scale from a likely predefined value), a green fixation cross was presented; if not, a red fixation cross was presented, and feedback was provided.

While performing the experiment, participants lay supine in a magnetically shielded room while continuous MEG data were recorded through a 157-channel whole-head axial gradiometer system (Kanazawa Institute of Technology, Kanazawa, Japan). Sampling rate was 1,000 Hz, and online high-pass filter of 1 Hz and low-pass filter of 200 Hz were applied. Five electromagnetic coils were attached to the forehead of the participants and their position was measured

twice, before the first and after the last block. Instructions, visual stimuli, and visual feedback were back-projected onto a Plexiglas screen using a Hitachi projector. Stimuli were presented using Psychtoolbox v3 ([88]; www.psychtoolbox.org), running under MATLAB R2019a (MathWorks) on an Apple iMac model 10.12.6. Participants responded to the yes/no question with their index finger of their left and right hand, using a keypad. For each trial, we also collected feedback scores and RTs. The overall duration of the MEG experiment was approximately 60 minutes.

## Data analysis

**Experiment 1 (and replication): RTs and mouse trajectories data.** The RTs and mouse trajectory analyses were limited to trials with positive feedback (group mean feedback scores: 82%, SD: 13%), and RTs were limited within the range of participant median RTs ± 2 SD.

To evaluate differences in RTs between antonyms ("small," "cold," "ugly," "bad," "sad" versus "big," "hot," "beautiful," "good," "happy," "fast," i.e., *low* versus *high* poles in each scalar dimension), and between negated and affirmative phrases (e.g., "really really good" versus "really not good"), and their interactions, median RTs of each participant were entered into 2 (*antonym*: low versus high) × 2 (*negation*: negated versus affirmative) repeated-measures ANOVA.

To evaluate differences in the final interpretations between antonyms in each scale, between negated and affirmative phrases, and their interactions, mean and standard deviation of the final responses of each participant were entered into a 2 (*antonym*: low versus high) × 2 (*negation*: negated versus affirmative) repeated-measures ANOVA. Post hoc tests were conducted for significant interactions (correction = Holm). Effect sizes were calculated using partial eta squared ($\eta_p^2$).

To compare mouse trajectories over time across participants, we resampled participants' mouse trajectories at 100 Hz using linear interpolation, up to 2 seconds, to obtain 200 time points for each trial. Furthermore, trajectories were normalized between −1 and 1. For visualization purposes, we computed the median of trajectories across trials for each participant, dimension (e.g., quality), antonym (e.g., "bad"), and modifier (e.g., "really not"), and at each time point.

Finally, to quantitatively evaluate how the interpretation of each phrase changed over time, for every participant, we carried out regression analyses per each time point, for affirmative and negated phrases separately (for a similar approach, see [45]). Note that, for the replication of Experiment 1, trials with "not not" were not included in this analysis, as the trajectories pattern was different compared to the other conditions with negation. The dependent variable was the mouse coordinate along the scale (the scale which was swapped in half of the trials was swapped back for data analysis purposes), and the predictor was whether the adjective was a low or high antonym (e.g., "bad" versus "good"). To identify the time windows where predictors were significantly different from 0 at the group level, we performed permutation cluster tests on beta values (10,000 permutations) in the time window from the onset of the adjective up to 1.4 seconds from adjective onset (i.e., 2 seconds from the onset of word 1).

**Experiment 2: Feedback scores and RTs data.** To evaluate differences in feedback scores between *low* and *high* antonyms ("bad," "cool," "quiet," "dark" versus "good," "warm," "loud," "bright"), and between negated and affirmative phrases (e.g., "really really good" versus "really not good"), and their interactions, mean feedback score in the yes/no task of each participant, computed as an average of 0 (red cross) and 1 (green cross) were entered into 2 (*antonym*: low versus high) × 2 (*negation*: negated versus affirmative) repeated-measures ANOVA.

The response time analysis was limited to trials with positive feedback. RTs outside the range of participant median RTs ± 2 SD were removed. To evaluate differences in RTs between

*low* and *high* antonyms in each scale and between negated and affirmative phrases, and their interactions, median RTs of each participant in the yes/no task were entered into a 2 (*antonym*: low versus high) × 2 (*negation*: negated versus affirmative) repeated-measures ANOVA.

**Experiment 2: MEG data.**

**Preprocessing.** MEG data preprocessing was performed using MNE-python [89] and Eelbrain (10.5281/zenodo.438193). First, bad channels (i.e., below the third or above the 97th percentile across all channels, for more than 20% of the entire recording) were interpolated. The MEG responses were denoised by applying least square projections of the reference channels and removing the corresponding components from the data [90]. Denoised data were lowpass-filtered at 20 Hz for the decoding analyses and at 40 Hz for the time-frequency analyses. FastICA was used to decompose the signal into 20 independent components, to visually inspect and remove artifacts related to eye-blinks, heartbeat, and external noise sources (removed components across blocks and participants: mean = 5.98, SD = 1.73). MEG recordings were then epoched into epochs of −300 ms and 2,550 ms around the onset of the first, second, or third word (or probe) for the decoding analyses, and into epochs of −800 and 3,000 ms around the onset of the first word for the time-frequency analyses (and then cut between −300 and 2,550 ms for group analyses). Note that, for visualization purposes, only 1,700 ms from the onset of the first word (i.e., 1,000 ms from adjective onset) were included in most figures (as no significant results were observed for control analyses run for later time windows). Finally, epochs with amplitudes greater than an absolute threshold of 3,000 fT were removed and a baseline between −300 to 0 ms was applied to all epochs.

**Source reconstruction.** Structural magnetic resonance images (MRIs) were collected for 10 out of 26 participants. For the remaining 16 participants, we manually scaled and coregistered the "fsaverage" brain to the participant's head-digitized shape and fiducials [89,91].

For every participant, an ico-4 source space was computed, containing 2,562 vertices per hemisphere and the forward solution was calculated using the Boundary Element Model (BEM). A noise covariance matrix was estimated from the 300 ms before the onset of the first word up to the onset of the first word presentation. The inverse operator was created and applied to the neuromagnetic data to estimate the source time courses at each vertex using dynamic statistical parametric mapping (dSPM; [92]). The results were then morphed to the ico-5 "fsaverage" brain, yielding to time courses for 10,242 vertices per hemisphere. We then estimated the magnitude of the activity at each vertex (signal to noise ratio: 3, lambda2: 0.11, with orientation perpendicular to the cortical surface), which was used in the decoding analyses (*Spatial decoders*).

**Decoding analyses.** Decoding analyses were limited to trials with positive feedback and were performed with the MNE [89] and Scikit-Learn packages [48]. First, X (or the selected principal components) were set to have 0 mean and unit variance (i.e., using a standard scaler). Second, we fitted an l2-regularized logistic regression model as estimator to a subset of the epochs (training set, $X_{train}$) and estimated y on a separate group of epochs (test set, $\hat{y}_{test}$). We then computed the accuracy (AUC; see below) of the decoder, by comparing $\hat{y}_{test}$ with the ground truth y. For this analysis, we used the default values provided by the Scikit-Learn package and set the class-weight parameter to "balanced."

**Temporal decoders.** Temporal decoding analyses were performed in sensor-space. Before fitting the estimators, linear dimensionality reduction (principal component analysis (PCA)) was performed on the channel amplitudes to project them to a lower dimensional space (i.e., to new virtual channels that explained more than 99% of the feature variance). We then fitted the estimator on each participant separately, across all selected components, at each time point separately. Time was subsampled to 100 Hz. We then employed a 5-fold (for analyses in **Fig 4B** and **4C**) or 10-fold stratified cross-validation (for analyses in **Figs 5A, 5C, and 6C**) that

fitted the estimator to 80% or 90% of the epochs and generated predictions on 20% or 10% of the epochs, while keeping the distributions of the training and test set maximally homogeneous. To investigate whether the representation of antonyms was comparable between affirmative and negated phrases, in a different set of analyses (i.e., decoding approach (i), **Fig 6B**), we fitted the estimator to all epochs corresponding to affirmative phrases and generated predictions on all epochs corresponding to negated phrases. In both decoding approaches, accuracy and probability estimates for each class were then computed. Decoding accuracy is summarized with an empirical area under the curve (rocAUC, 0 to 1, chance at 0.5).

At the group level, we extracted the clusters of time where AUC across participants was significantly higher than chance using a one-sample permutation cluster test, as implemented in MNE-python (10,000 permutations [93]). We performed separate permutation cluster tests for the following time windows: −700 to −350 ms from adjective onset (i.e., word 1), −350 to 0 ms from adjective onset (i.e., word 2), 0 to 500 ms from adjective onset (i.e., time window for lexical-semantic processes [65,66]), and 500 to 1,000 ms from adjective onset (i.e., to account for potential later processes).

**Expected outcome for the effect of negation on the representation of antonyms.** Temporal decoding approach (i) and (ii) described above allow us to make specific predictions about the effect of negation on the representation of antonyms (**Fig 6A**).

*Approach (i)* train set: affirmative phrases (in green in **S2 Table**); test set: negated phrases (in red in **S2 Table**). For our results to support predictions (1) *No effect* or (2) *Mitigation*, this decoding approach should show probability estimates of high and low adjectives significantly above the computed chance level and in the direction of the respective classes, indicating that the initial representation of adjectives in negated phrases is similar to that in affirmative phrases (left column, first and second row under *decoding approach* in **Fig 6A**). Conversely, for our results to support prediction (3) *Inversion*, this decoding approach should show probability estimates of high and low adjectives significantly above the computed chance level but in the direction of the opposite classes (i.e., swapped), as adjective representations would be systematically inverted in negated phrases (left column, third row under *decoding approach* in **Fig 6A**). Finally, we should observe at chance probability estimates in the case of (4) *Change*, where adjective representations in negated phrases are not predictable from the corresponding representations in affirmative phrases (left column, fourth row under *decoding approach* in **Fig 6A**).

*Approach (ii)* train set: affirmative and negated phrases together (in green/red in **S2 Table**); test set: affirmative and negated phrases separately (in green and red in **S2 Table**). This decoding analysis allows us to disentangle predictions (1) *No effect* from (2) *Mitigation*. For the results of this analysis to support prediction (1) *No effect*, we should observe quantitatively comparable probability estimates in affirmative and negated phrases, suggesting that negation does not change the representation of adjectives (right column, first row under *decoding approach* in **Fig 6A**). Conversely, in support of prediction (2) *Mitigation*, we should observe significantly reduced (or even at chance) probability estimates for negated relative to affirmative phrases, suggesting less robust differences between low and high antonyms in negated phrases (right column, second row under *decoding approach* in **Fig 6A**). The outcome of predictions (3) *Inversion* would be at chance probability estimates for affirmative and negated phrases (as the model is trained on opposite representations within the same class; right column, third row under *decoding approach* in **Fig 6A**) and the outcome of (4) *Change* would be at chance probability estimates for negated phrases (as the model is trained on different representations within the same class; right column, fourth row under *decoding approach* in **Fig 6A**).

**Spatial decoders.** Spatial decoding analyses were performed in source-space. We fitted each estimator on each participant separately, across 50 to 650 ms time samples relative to the onset of the adjective (to include the 3 significant time windows that emerge from the temporal decoding analysis in **Fig 4B**), at each brain source separately, after morphing individual participant's source estimates to the ico-5 "fsaverage" common reference space. We employed a 5-fold stratified cross-validation, which fitted the estimator to 80% of the epochs and generated predictions on 20% of the epochs, while keeping the distributions of the training and test set maximally homogeneous. Decoding accuracy is summarized with an empirical area under the curve (AUC, 0 to 1, chance at 0.5). At the group level, we extracted the brain areas where the AUC across participants was significantly higher than chance, using a 1-sample permutation cluster test as implemented in MNE-python (10,000 permutations; adjacency computed from the "fsaverage" brain [93]).

**Time-frequency analysis.** We extracted time-frequency power of the epochs (−800 to 3,000 ms from the onset of word 1) using Morlet wavelets of 3 cycles per frequency, in frequencies between 3.9 and 37.2 Hz, logarithmically spaced (19 frequencies overall). Power estimates were then cut between −300 and 2,550 ms from onset of word 1 and baseline corrected using a window of −300 to −100 ms from the onset of word 1, by subtracting the mean of baseline values and dividing by the mean of baseline values (mode = "percent"). Power in the low-beta frequency range (12 to 20 Hz) and in the high-beta frequency range (21 to 30 Hz [57,79]) was averaged to obtain a time course of power in low and high-beta rhythms. We then subtracted the beta power of affirmative phrases from that of negated phrases. At the group level, we extracted the clusters of time where this difference in power across participants was significantly greater than 0, using a 1-sample permutation cluster test as implemented in MNE-python (10,000 permutations [93]). We performed separate permutation cluster tests in the same time windows used for the decoding analysis: −700 to −350 ms, −350 to 0 ms, 0 to 500 ms, and 500 to 1,000 ms from the onset of the adjective (note that no significant differences were observed in analyses ran for time windows after 1,000 ms). We then computed the induced power in source space (method: dSPM and morphing individual participant's source estimates to the ico-5 "fsaverage" reference space) for the significant clusters of time in the low- and high-beta range separately and averaged over time. At the group level, we extracted the brain areas where the power difference across participants was significantly greater than 0, using a 1-sample permutation cluster test as implemented in MNE-python (10,000 permutations; adjacency computed from the "fsaverage" brain [93]).

## Supporting information

**S1 Fig. Trajectories for each scalar dimension in Experiment 1.** Behavioral trajectories for low (purples) and high (oranges) antonyms over time, for each scalar dimension (i.e., quality, beauty, mood, temperature, speed, and size), for each modifier (shades of orange and purple), and for affirmative and negated phrases. Black vertical dashed lines indicate the presentation onset of each word: modifier1, modifier2, and adjective.
(TIF)

**S2 Fig. Temporal decoding of negation as a function of number of modifiers (i.e., complexity), time-locked to the onset of the probe.** Decoding accuracy of negation over time, as a function of the number of modifiers (1 modifier: dark red line and shading; 2 modifiers: light red line and shading). Significant time windows are indicated by dark red (1 modifier) and light red (2 modifiers) shading. These results show that we could significantly decode the difference between affirmative and negated phrases between 230 and 930 ms after the onset of the probe, especially when the phrase included two modifiers (1 modifier: between 790 and 930

ms: $p < 0.001$; 2 modifiers: between 230 and 840 ms: $p < 0.001$). This suggests that the representation of modifiers is reactivated at the stage when participants have to perform the yes/no task. 1 modifier: "really ###," "### really," "not ###," "### not"; 2 modifiers: "really really," "really not," "not really," "not not." AUC = area under the receiver operating characteristic curve, chance = 0.5 (black dashed horizontal line); the black vertical dashed line indicates the presentation onset of the probe; aff = affirmative, neg = negated; each line and shading represent participants mean ± SEM.
(TIF)

**S3 Fig. Temporal decoding of composed meaning.** We trained estimators on phrases where the predicted composed meaning was "low" vs. "high" in 90% of the trials and computed the accuracy of the model in predicting the representation of the meaning "low" vs. "high" in the remaining 10% of the trials. For example, for the *quality* dimension, classes are: [0: *bad*] "### really bad," "really ### bad," "really really bad," "### not good," "not ### good," "not not good," "really not good," "not really good"; and [1: *good*] "### really good," "really ### good," "really really good," "### not bad," "not ### bad," "not not bad," "really not bad," "not really bad." The composed meaning was derived from the behavioral results of Experiment 1. (**A**) Temporal decoding analyses time-locked to the onset of the adjective do not reveal any significant temporal cluster, suggesting that negation does not invert the representation of the adjective to that of its antonym (e.g., "bad" to "good"), as would be predicted by prediction (3) *Inversion*. (**B**) Temporal decoding analyses time-locked to the onset of the probe do not reveal any significant temporal cluster. For all panels: AUC = area under the receiver operating characteristic curve, chance = 0.5 (black horizontal dashed line); black vertical dashed lines indicate the presentation onset of the adjective in **A** and the probe in **B**; each line and shading represent participants' mean ± SEM.
(TIF)

**S4 Fig. Follow-up analyses of Fig 6C.** (**A**) We conducted a follow-up analysis where we trained and tested on "low" vs. "high" antonyms in affirmative and negated phrases separately, to further investigate lowering in decoding accuracy when representations are closer on the semantic scale, as predicted by the mitigation hypothesis for negated phrases. We found similar patterns to our main analysis. Results show that affirmative phrases (green line) are associated with significantly above-chance decoding accuracy between 150 and 190 ms ($p = 0.026$; green shading and horizontal solid line) from adjective onset. No significant above-chance decoding accuracy was found for negated phrases before approximately 400 ms from adjective onset (390 to 440 ms, $p = 0.009$; red shading and horizontal solid line). (**B**) We conducted a follow-up analysis where no trials were removed due to the feedback score. We found similar patterns to our main analysis. Results show that affirmative phrases (green line) are associated with significantly above-chance decoding accuracy between 100 and 190 ms and 230 and 280 ms from adjective onset ($p = 0.001$ and $p = 0.032$, respectively, green shading and horizontal solid lines). Negative phrases (red line) are associated with significantly above-chance decoding accuracy between 350 to 440 ms from adjective onset ($p < 0.001$, red shading and horizontal solid line). (**C–E**) We conducted a series of follow-up analyses where we removed one condition (i.e., 1 modifiers combination) at a time to evaluate its specific effect on adjective representation. (**C**) "not not" is removed from the analysis: Affirmative phrases (green line) are associated with significantly above-chance decoding accuracy between 130 and 280 ms from adjective onset ($p < 0.001$, green shading and horizontal solid line); negative phrases (red line) are associated with significantly above-chance decoding accuracy between 200 to 250 ms and between 380 to 430 ms from adjective onset ($p = 0.011$ and $p = 0.049$, red shading and horizontal solid lines). (**D**) "really not" is removed from the analysis: Affirmative phrases (green line) are associated with significantly above-chance decoding accuracy between 140 and 280

ms and between 370 and 420 ms from adjective onset ($p = 0.001$ and $p = 0.038$, green shading and horizontal solid lines); negative phrases (red line) are associated with significantly above-chance decoding accuracy between 190 to 260 ms from adjective onset ($p = 0.009$, red shading and horizontal solid lines). (**E**) "really really" is removed from the analysis: Affirmative phrases (green line) are associated with significantly above-chance decoding accuracy between 150 and 190 ms from adjective onset ($p = 0.025$, green shading and horizontal solid line); no significant above-chance decoding accuracy was found for negated phrases. Overall, these results suggest that "not not" and "really not" have similar mitigation effects. Conversely, and as expected, "really really" does not have mitigation effects on adjective representation.
(TIF)

**S5 Fig. Differences between negated and affirmative phrases across time and frequencies.** Time-frequency spectrum of the differences between negated and affirmative phrases averaged across all sensors and all participants. Frequencies are between 3.9 and 37.2 Hz, logarithmically spaced. Black vertical dashed lines indicate the presentation onset of each word: modifier1, modifier2, and adjective; colors indicate % differences in change relative to a baseline of −300 to −100 ms from the onset of word 1 (modifier1).
(TIF)

**S6 Fig. Low- and high-beta power for negated and affirmative phrases across time.** The mean beta power for the no modifier condition was subtracted from the mean beta power of affirmative and negated phrases, separately for low-beta (12–20 Hz, (**A**)) and high-beta (21–30 Hz, (**B**)). The horizontal solid black line represents the no modifier condition (i.e., ### ###) after subtraction (thus = 0), and the green and red lines represent beta power over time for affirmative and negated phrases, respectively. Relative change (%) was obtained by subtracting the mean of baseline values (−300 to −100 ms from the onset of word1) and dividing by the mean of baseline values. Black vertical dashed lines indicate the presentation onset of each word: modifier1, modifier2, and adjective; each line and shading represent participants' mean ± SEM.
(TIF)

**S1 Table. Comprehensive list of the 108 stimuli used in the behavioral experiment (and replication), color coded for each experimental condition; purple: low adjectives, orange: high adjectives; green: affirmative phrases, red: negated phrases.**
(DOCX)

**S2 Table. Comprehensive list of the 72 stimuli used in the MEG experiment, color coded for each experimental condition; purple: low adjectives, orange: high adjectives; green: affirmative phrases, red: negated phrases.** Note that the condition with no modifiers ("### ###") was only employed as a baseline condition in the time-frequency analysis.
(DOCX)

**S3 Table. We performed a one-way ANOVA and Tukey post hoc tests on the average RTs across trials per participant and per each modifier condition.** Each line represents a pairwise comparison between each pair of modifiers, for Experiment 1 (i.e., behavioral experiment, **A**) and its replication (**B**). *p*-Value and confidence intervals are adjusted for comparing a family of 9 estimates. Significant *p*-values are highlighted in bold.
(DOCX)

## Author Contributions

**Conceptualization:** Arianna Zuanazzi, Pablo Ripollés, Jean-Rémi King, David Poeppel.

**Data curation:** Arianna Zuanazzi, Pablo Ripollés, Wy Ming Lin.

**Formal analysis:** Arianna Zuanazzi, Pablo Ripollés, Jean-Rémi King.

**Funding acquisition:** David Poeppel.

**Investigation:** Arianna Zuanazzi, Pablo Ripollés, Jean-Rémi King, David Poeppel.

**Methodology:** Arianna Zuanazzi, Pablo Ripollés, Laura Gwilliams, Jean-Rémi King.

**Project administration:** Arianna Zuanazzi.

**Resources:** David Poeppel.

**Supervision:** Pablo Ripollés, Jean-Rémi King, David Poeppel.

**Validation:** Arianna Zuanazzi.

**Visualization:** Arianna Zuanazzi.

**Writing – original draft:** Arianna Zuanazzi.

**Writing – review & editing:** Arianna Zuanazzi, Pablo Ripollés, Laura Gwilliams, Jean-Rémi King, David Poeppel.

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
