## [Editor Report · Decision Letter 0]

9 Aug 2023

Dear Dr Zuanazzi, 

Thank you for submitting your manuscript entitled "Tracking the behavioral and neural dynamics of semantic representations through negation" for consideration as a Research Article by PLOS Biology.

Your manuscript has now been evaluated by the PLOS Biology editorial staff as well as by an academic editor with relevant expertise and I am writing to let you know that we would like to send your submission out for external peer review.

Once your full submission is complete, your paper will undergo a series of checks in preparation for peer review. After your manuscript has passed the checks it will be sent out for review. To provide the metadata for your submission, please Login to Editorial Manager (https://www.editorialmanager.com/pbiology) within two working days, i.e. by Aug 11 2023 11:59PM.

Kind regards,

Christian

Christian Schnell, PhD

Senior Editor

PLOS Biology

cschnell@plos.org

---

## [Decision Letter · Decision Letter 1]

26 Sep 2023

Dear Dr Zuanazzi,

Thank you for your patience while your manuscript "Tracking the behavioral and neural dynamics of semantic representations through negation" was peer-reviewed at PLOS Biology. It has now been evaluated by the PLOS Biology editors, an Academic Editor with relevant expertise, and by several independent reviewers. 

In light of the reviews, which you will find at the end of this email, we would like to invite you to revise the work to thoroughly address the reviewers' reports.

As you will see below, the reviewers think that the study is overall well executed and provides important insights. However, Reviewer 1 has some concerns about the consistency between MEG and behavioural findings, and also a couple of technical and conceptual concerns. Reviewer 2 also raises conceptual concerns, while Reviewer 3 is the most positive reviewer and has only relatively minor concerns.

After discussing the reports with the Academic Editor, who agrees with Reviewer 2' conceptual concerns, we encourage you take the opportunity to further clarify and organize the conceptual and theoretical claims when revising your manuscript. This would include to refrain from interpreting the timecourse and strength of decoding effects so directly, better developing the idea/theory that negation would result in an inverse of the neural response or in decoding performance, and better fleshing out the idea of mitigation (conceptually and ideally formally).

Given the extent of revision needed, we cannot make a decision about publication until we have seen the revised manuscript and your response to the reviewers' comments. Your revised manuscript is likely to be sent for further evaluation by all or a subset of the reviewers.

**IMPORTANT - SUBMITTING YOUR REVISION**

*Re-submission Checklist*

*Published Peer Review*

*PLOS Data Policy*

*Blot and Gel Data Policy*

Sincerely,

Christian

Christian Schnell, PhD

Senior Editor

PLOS Biology

cschnell@plos.org

REVIEWS:

Reviewer #1: This manuscript investigates the time course of negated phrases in an abstract semantic meaning context. The authors used a behavioral paradigm and an MEG study to show that negation does not simply invert the meaning of the word, but rather mitigates the meaning of a words. This study used a smart paradigm and combined this with nice decoding approaches. I have some questions and remarks.

Questions regarding the consistency of the MEG and the behavioral findings.

1) In the behavioral findings you show that participant first interpret the word as written and only later revert the meaning. Was it investigated whether the representations in MEG show the same pattern? If this pattern couldn't be found, how do the authors interpret this?

2) The behavioral findings of final word interpretation don't really speak in favor for mitigation, but rather are in line with the inversion (not a full inversion, but still inverted). How do the authors connect the finding of the behavior with the MEG?

Question on the decoding approach

3) Hypothesis figure in 5. Why for the change interpretation wouldn't one be able to decode in analysis ii) the affirmative conditions anymore? In my opinion the 'changed' representation would simply add noise to the training data set. Therefore, you could still decode the affirmative (maybe less strongly though).

4) In all the decoding approaches for low versus high there are only 4 words in the low and 4 words in the high category. It is possible that some of the word-specific, potentially visual features end up in the decoding rather than having it based on valence? This promotes the decoding of the negated to be in-line with the affirmative option which might only related to irrelevant features. This will make it more difficult to find anything in line with the inversion hypothesis (which is more what the behavior speaks for). To get around this, I would train on three of the word pairs and test on the last word pair (and cross-validate). In this way one is sure that the effects are not due to any early sensory processing. 

5) In the mitigation hypothesis, one would also expect that when tested and trained on affirmative condition the decoding should work better than when tested and trained on negated condition. This occurs because it is more difficult to decode when the features are closer in some abstract multi-dimensional space (which is assume for the mitigation). I wondered why this analysis was not pursued as it seems the most simple one.

6) If you believe that there is a linear space on which the features can be mapped, why not approach the decoding as a regression rather than a binary feature. Then you could map everything in one go.

Questions on design/method choices

7) Why are the words for the MEG and behavior different?

8) Wouldn't it make sense to also analyze the data taking the 'not not' out? I understand you had it original because of the factorial design, but if for the participants it is so different one cannot really treat is as such (as also supplementary figure 2 shows).

9) Why was decoding limited to only correct trials? This might force the analysis to only be focused on trials where the decoding was in line with the experimentally induced representation of the negated items. 

Questions on interpretation

10) In my opinion the 'replication' which is now the supplementary should be the main figure of the behavioral paper. If feedback was given what is the right answer in the original study you did not show how negation is represented, merely how participants can learn to follow instructions. It is good that the main patterns replicate, but I feel that now it is deceiving to put the effect so strongly as they appear in figure 1.

11) I don't really see how you can define accuracy here. Depending on the context 'not bad' could feel closer to 'good' or closer to 'bad'. So, given that here there is no context how can you ever provide a definition of accuracy? This is a problem for the original behavioral study (especially as conclusion are drawn based on this). But it is the same issue with the MEG. How can you provide feedback on a 1-8 scale on how to interpret these abstract phrases? I also wonder whether this feedback forces people to represent the information in a specific manner that is forced by the experimenter. In the behavioral data that is now checked, but in the MEG not so this could bias the decoding.

12) Is beta necessarily inhibition or could it also be something else? Beta has also been implied generally for feedback, not necessary for negative feedback.

13) In the conclusion it is stated that the behavioral effect shows that the negation never inverts the meaning of the word. While I agree it shows that it never completely inverts the meaning, one cannot state there is no inversion at all. It almost seems to suggest that the behavioral effects of the final decision support the MEG results, which I find quite a strong interpretation gives that in the MEG the negation does not cause a negated interpretation, but in the behavior it does. I think the trajectories have a good story that the initial representation matches the affirmative adjective, but for the final decision I think it simply doesn't match.

Other

14) In the abstract a conclusion is made on the MEG, but it would be nice to spend just a few words on how this conclusion could be reached.

15) Line 671. What does the swapping of yes/no means. Is it just the response buttons or something else? This wasn't clear.

Reviewer #2: The article tests how combinatoric representation of negation and adjectives operates in both behavioral and MEG experiments. The results from RT and mouse trajectory show that negated adjectives do not change into the opposite meaning (e.g. "not good" doesn't become "bad".). The MEG decoding analysis shows that negation mitigates the meaning of antonyms, ruling out other three possible hypotheses (i.e., no change, inversion, change). The authors also report that beta power was increased in negation, in line with the suppression effect in the previous research. Overall, I found the research was conducted in a rigorous way. It is also valuable to see what kind of combinatoric processes relating to negation occur in our brain. However, the overall analysis is unclear to address the research question. Please find below for my comments. 

Major comments: I was confused when I read the plots since it's not really straightforward and clear what conditions the authors compared. The authors had an elegant design of the experiment. However, it is not clear to me how they compare the conditions they have. For example, in Fig.4C, they showed decoding accuracy of negation with 1 and 2 modifiers. "1 modifier" could mean "### really good", "### not good", "really ### good", "not ### good". Maybe also including an example explicitly in the plot might make it clearer. Also, I am concerned about the effect of placeholder "###". For my intuition, it would be weird to read the stimuli like "really ### good". I think the authors should have some additional analysis to rule out any potential effect from the placeholder. For example, should there be no difference when comparing "### really" vs. "really ###" if the placeholder does not affect the combinatoric processes? The overall article may mainly focus on the comparison "really not good" vs. "really really good", but why not just compare "really good" vs. "not good"? I was not really sure if the experiments really needed stimuli with placeholder if they didn't show full analysis on those stimuli. Since the authors have included stimuli with "###", I think the authors should also analyze them or specify them explicitly in the text/plots. 

Overall, it is sometimes not straightforward how some analyses address the research questions. For example, how does the analysis in Fig. 4A and 4B address the neural processes of negation? It seems like the comprehensive design is used somewhat incompletely in terms of the contrasts used in certain analyses (when pooling over different modifier conditions. In general, the steps between questions, hypotheses, predictions (of the results) and analyses could be more clearer. 

Minor comments: 

Introduction:

Ln 72-75: It's not clear to me here what you mean: "the building blocks" and "stem from more subtle inferential meaning". The examples given here could be explained further. 

Ln 119: why is the decoding approach applied? 

Results/Methods

Exp1

Reaction times

Ln 170-171: Participants were faster for affirmative phrases (e.g. really really good) than for negated phrases (e.g. really not good). What about "not really good" and also "really good" vs. "not good"? I also wonder if there is a difference in "### really" vs. "really ###" and "### not" vs. "not ###". I would expect no difference here. It would be great if the authors could provide pairwise comparisons here. 

Exp 2

Ln 736: Please include the interval of the number of removed components.

Ln 752: Why was the noise covariance matrix estimated from the 300ms before the onset of the first word? And up to when? 

Ln 774-775: So does it mean that there is a threshold for the number of trials to decide when you use 5-fold/10-fold cross-validation? Is it possible to report a number here? 

Ln 796/Ln 809 / Ln 351: maybe state explicitly what train/test set were used for both decoding approaches. The authors only mentioned train/test for the first approach, but not the second, though I assume the train set would include "really really bad/good" and "really not bad/good". 

Ln 817-819/Fig. 5A: Though it looks like the authors provided some kind of explanation here, I am still not clear why the prediction of (3) inversion would have the same prediction of (4)? Why not expect a reversed direction for negation? 

Fig.3: overall, I am not sure if these really reflect single word processing for modifiers. The authors might need to include control analysis of isolated adjective ("### ### good") and also ba

---

## [Decision Letter · Decision Letter 2]

25 Jan 2024

Dear Dr Zuanazzi,

Thank you for your patience while we considered your revised manuscript "Tracking the behavioral and neural dynamics of semantic representations through negation" for consideration as a Research Article at PLOS Biology. Your revised study has now been evaluated by the PLOS Biology editors, the Academic Editor and two of the original reviewers. 

In light of the reviews, which you will find at the end of this email, we are pleased to offer you the opportunity to address the comments from Reviewer 1 and the Academic Editor in a revision that we anticipate should not take you very long. 

Specifically, after discussing your response to Reviewer 2's concerns (who was not available to review the revised manuscript) with the Academic Editor, we feel that some of the limitations that were mentioned by Reviewer 2 (and partially by Reviewer 3) in the previous round of review should be discussed more clearly. Therefore, we would ask you to:

* address Reviewer 1's remaining points

* discuss the limitations of the study design and interpretation of the results more clearly

* deepen the discussion of the theoretical insights your study can offer (as suggested by Reviewer 2 and 3 in the previous round of review)

We will then assess your revised manuscript and your response to the reviewers' comments with our Academic Editor aiming to avoid further rounds of peer-review, although might need to consult with the reviewers, depending on the nature of the revisions.

**IMPORTANT - SUBMITTING YOUR REVISION**

*Resubmission Checklist*

*Published Peer Review*

*PLOS Data Policy*

*Blot and Gel Data Policy*

Sincerely,

Christian

Christian Schnell, PhD

Senior Editor

PLOS Biology

cschnell@plos.org

REVIEWS:

Reviewer #1: I thank the reviewers for their elaborate responses. Many of my points have been addressed. I do still have a few remaining point that need addressing. 

1) The authors try to link the MEG and the behavioral data. Indeed, I agree that at the early stages of the behavioral choices the participants for the negated options divert first to the other response option (which could be indicative of mitigation). However, to better link the behavioral data to the MEG data, it would be good to show that at the decision stage there is an inversion in the MEG responses as well. This can be done by locking the data to the probe or to the response and repeating the decoding analysis. Especially considering that they have reverted the response options there is no confound of button presses here.

2) I do not feel that the authors acknowledge the problem of visual similarity ending up in the decoding sufficiently. While I acknowledge that their blocked design makes it difficult to do this analysis, I still hold that if visual features influence the decoding, it is difficult to differentiate the outcome expectations for the different models. For example, see figure 6. Let us say that the true underlying representation is a changed representation (4), but part of the decoding is due to visual differences. That would effectively increase the overall decoding (as the sensory representations are the same). This increase would lead exactly to a pattern consistent with mitigation. I do not think they can currently differentiate these options. 

Reviewer #3: I read the revised manuscript and the authors' rebuttal to my comments, which were satisfactorily addressed. I have no further comments and so endorse this manuscript for publication.

---

## [Decision Letter · Decision Letter 3]

19 Mar 2024

Dear Dr Zuanazzi,

Thank you for your patience while we considered your revised manuscript "Tracking the behavioral and neural dynamics of semantic representations through negation" for publication as a Research Article at PLOS Biology. This revised version of your manuscript has been evaluated by the PLOS Biology editors, the Academic Editor and the original reviewers.

Based on the reviews and on our Academic Editor's assessment of your revision, we are likely to accept this manuscript for publication, provided you satisfactorily address the remaining points raised. 

In particular, we would like your text to reflect that without quantification or control of perceptual processes and their contribution to decoding, the situation Reviewer 1 described in the previous round of review could occur; and we would like to make this point clearer to the reader in the discussion. One suggestion would be:

"However, we note that if the consistency or probability of visual features influences the decoding performance, discriminating between the predicted patterns for our competing hypotheses becomes more difficult. For example, if the underlying representation is a changed representation (see 4 in Fig.6), but part of the decoding performance is attributable to visual consistencies, that would effectively increase the overall decoding performance, leading to a pattern consistent with mitigation (see 2 in Fig.6)."

Please note that we offer this text as a guideline, and you are, of course, are free to adapt it.

Please also make sure to address the following data and other policy-related requests.

* We would like to suggest a different title to improve readability for our broad audience: "Negation mitigates rather than inverts the neural representations of adjectives"

* Please add the links to the funding agencies in the Financial Disclosure statement in the manuscript details.

* Please include information about the form of consent (written/oral) given for research involving human participants. All research involving human participants must have been approved by the authors' Institutional Review Board (IRB) or an equivalent committee, and must have been conducted according to the principles expressed in the Declaration of Helsinki.

DATA POLICY:

Regardless of the method selected, please ensure that you provide the individual numerical values that underlie the summary data displayed in the following figure panels as they are essential for readers to assess your analysis and to reproduce it: 2A, 3A, 6D, and 6E

CODE POLICY

Per journal policy, as the code that you have generated is important to support the conclusions of your manuscript, we require that you make it available without restrictions upon publication. Please ensure that the code is sufficiently well documented and reusable, and that your Data Statement in the Editorial Manager submission system accurately describes where your code can be found.

* Please make the data repository https://osf.io/5ys6b/ accessible, currently it is only accessible with username and password.

We expect to receive your revised manuscript within two weeks. 

*Published Peer Review History*

*Press*

Sincerely,

Christian

Christian Schnell, PhD, 

Senior Editor

cschnell@plos.org

PLOS Biology

Reviewer remarks:

Reviewer #1: I have no further comments to the authors and thank them for their reply.

---

## [Editor Report · Decision Letter 4]

11 Apr 2024

Dear Arianna,

Thank you for the submission of your revised Research Article "Negation mitigates rather than inverts the neural representations of adjectives" for publication in PLOS Biology. On behalf of my colleagues and the Academic Editor, Andrea Martin, I am pleased to say that we can in principle accept your manuscript for publication, provided you address any remaining formatting and reporting issues. These will be detailed in an email you should receive within 2-3 business days from our colleagues in the journal operations team; no action is required from you until then. Please note that we will not be able to formally accept your manuscript and schedule it for publication until you have completed any requested changes.

PRESS

Sincerely, 

Christian

Christian Schnell, PhD

Senior Editor

PLOS Biology

cschnell@plos.org